# Room-temperature sub-100 nm Néel-type skyrmions in non-stoichiometric van der Waals ferromagnet Fe$_{3-x}$GaTe$_2$ with ultrafast laser writability

Zefang Li [1,10], Huai Zhang [2,10], Guanqi Li [3,10], Jiangteng Guo[1], Qingping Wang[4], Ying Deng[1], Yue Hu[1], Xuange Hu[1], Can Liu[1], Minghui Qin [2], Xi Shen[5], Richeng Yu[5], Xingsen Gao [2], Zhimin Liao [6], Junming Liu [2,7], Zhipeng Hou [2] ✉, Yimei Zhu [8] ✉ & Xuewen Fu [1,9] ✉

Realizing room-temperature magnetic skyrmions in two-dimensional van der Waals ferromagnets offers unparalleled prospects for future spintronic applications. However, due to the intrinsic spin fluctuations that suppress atomic long-range magnetic order and the inherent inversion crystal symmetry that excludes the presence of the Dzyaloshinskii-Moriya interaction, achieving room-temperature skyrmions in 2D magnets remains a formidable challenge. In this study, we target room-temperature 2D magnet Fe$_3$GaTe$_2$ and unveil that the introduction of iron-deficient into this compound enables spatial inversion symmetry breaking, thus inducing a significant Dzyaloshinskii-Moriya interaction that brings about room-temperature Néel-type skyrmions with unprecedentedly small size. To further enhance the practical applications of this finding, we employ a homemade in-situ optical Lorentz transmission electron microscopy to demonstrate ultrafast writing of skyrmions in Fe$_{3-x}$GaTe$_2$ using a single femtosecond laser pulse. Our results manifest the Fe$_{3-x}$GaTe$_2$ as a promising building block for realizing skyrmion-based magneto-optical functionalities.

Magnetic skyrmions, which are topological swirling spin configurations stabilized by Dzyaloshinskii–Moriya interaction (DMI)[1,2], have garnered significant interest over the past decade because of their nanometric scale and exotic magnetoelectronic properties[3], such as the topological Hall effect (THE)[4], skyrmion Hall effect[5], and ultra-low current density for motion[6,7], making them ideal information carriers for future high-density and fast-speed data storage[8], quantum and neuromorphic computation[9,10]. Since the existence of

[1]Ultrafast Electron Microscopy Laboratory, The MOE Key Laboratory of Weak-Light Nonlinear Photonics, School of Physics, Nankai University, Tianjin, China. [2]Guangdong Provincial Key Laboratory of Optical Information Materials and Technology, Institute for Advanced Materials, South China Academy of Advanced Optoelectronics, South China Normal University, Guangzhou, China. [3]School of Integrated Circuits, Guangdong University of Technology, Guangzhou, China. [4]School of Physics and Electronic and Electrical Engineering, Aba Teachers University, Wenchuan, China. [5]Beijing National Laboratory for Condensed Matter Physics, Institute of Physics, Chinese Academy of Sciences, Beijing, China. [6]State Key Laboratory for Mesoscopic Physics and Frontiers Science Center for Nano-optoelectronics, School of Physics, Peking University, Beijing, China. [7]Laboratory of Solid State Microstructures and Innovation Center of Advanced Microstructures, Nanjing University, Nanjing, China. [8]Condensed Matter Physics and Materials Science Department, Brookhaven National Laboratory, Upton, New York, USA. [9]School of Materials Science and Engineering, Smart Sensing Interdisciplinary Science Center, Nankai University, Tianjin, China. [10]These authors contributed equally: Zefang Li, Huai Zhang, Guanqi Li. ✉e-mail: houzp@m.scnu.edu.cn; zhu@bnl.gov; xwfu@nankai.edu.cn

skyrmion crystal was first verified in helimagnet MnSi[11], various skyrmion-host three-dimensional (3D) bulk materials have been discovered with abundant magnetic and electronic features[12,13]. Compared to the 3D bulk ferromagnets, two-dimensional (2D) van der Waals (vdW) ferromagnets have inherent superiorities for practical applications in spintronic devices due to their unique atomic layered structure[14,15], such as long-range magnetic order down to atomic thickness[16,17], wide flexibility for stacking artificial heterostructures[18], high sensitivity to external field perturbations[19], and possible compatibility with modern integrated circuit process[20]. Therefore, exploring magnetic skyrmions with small size, especially at room temperature (RT), in 2D vdW ferromagnetic materials with facile tunability have become a focal point of magnetic and topological order of matters, as well as spintronic applications[21].

To realize RT magnetic skyrmions within the isolated 2D vdW ferromagnets, DMI induced by inversion crystal symmetry breaking and RT ferromagnetism are crucial prerequisites[22]. Nevertheless, for most of the intrinsic 2D vdW ferromagnets discovered hitherto, such as $Fe_3GeTe_2$[23], $Fe_5GeTe_2$[24], $CrGeTe_3$[25], and $CrI_3$[26], etc., on one hand, their natural centrosymmetric crystal structures exclude the DMI;[27] on the other hand, due to the strong Mermin–Wagner fluctuations that suppress the intrinsic magnetic order in the 2D limit[28], their Curie temperatures ($T_c$) are typically below RT. Therefore, on the way pursing RT magnetic skyrmions in 2D vdW ferromagnets, how to simultaneously improve $T_c$ and introduce DMI is an internationally recognized difficulty. Although several methods, such as elemental doping and substitution etc.[29,30], have shown the possibility to increase $T_c$ while breaking the centrosymmetric structure for creating DMI in intrinsic 2D vdW ferromagnets[31,32], the elemental controllability and raising the $T_c$ above RT are still challenging[33,34]. One better choice is to directly introduce DMI into intrinsic 2D vdW ferromagnets with $T_c$ above RT. The iron-based ternary telluride $Fe_3GaTe_2$[35] and chromium-based binary telluride $CrTe_2$[36] are the only two 2D vdW intrinsic magnets that exhibit RT ferromagnetism discovered hitherto, but merely the $Fe_3GaTe_2$ exhibits both above $T_c$ and large RT perpendicular magnetic anisotropy (PMA) up to the order of magnitudes of $10^5$ J/$m^3$[35]. Recently, a $Fe_3GaTe_2$-based magnetic tunnel junction (MTJ) achieved a large tunnel magnetoresistance (TMR) of 85% at RT[37], highlighting the great potential of $Fe_3GaTe_2$ for developing spintronic devices. However, the intrinsically inversion-symmetric crystal structure of $Fe_3GaTe_2$ possesses an inherent obstacle. Hitherto, the universal strategy that enables directly breaking the inversion symmetry and achieving substantial DMI for stabilizing RT skyrmions in the $Fe_3GaTe_2$ have not been realized.

In this study, we discover that the iron deficiency in $Fe_3GaTe_2$ can lead to a pronounced displacement of the Fe atoms within the crystal structure. Based on systematic structural analysis and first-principles calculations, we find that this atomic displacement causes a transformation from the original centrosymmetric crystal structure to a non-centrosymmetric structure, resulting in a significant DMI. Combined Lorentz transmission electron microscopy (LTEM), magnetic force microscopy (MFM) and magneto-transport measurements demonstrate that the non-stoichiometric $Fe_{3-x}GaTe_2$ could accommodate Néel-type skyrmions together with a prominent topological Hall effect over a broad temperature range of 330 K to 100 K. Moreover, the size of the skyrmions decreases as the sample thickness becomes thinner, and field-free sub-100 nm skyrmions can be obtained at RT when the thickness falls below a threshold ranging from 40 to 60 nm, which are the smallest skyrmions achieved hitherto in 2D vdW magnets. More intriguingly, with the use of a homemade in-situ optical LTEM, we realize an ultrafast writing of RT skyrmions in the non-stoichiometric $Fe_{3-x}GaTe_2$ thin flakes by a single femtosecond (fs) laser pulse, which offers a possible avenue for the realization of ultrafast and energy-efficient skyrmion-based logic and memory devices.

## Results and discussions

To control the Fe content, we systematically grew a series of $Fe_{3-x}GaTe_2$ single crystals by varying the Fe content in the raw material composition, utilizing a Te-flux method (see Methods section and Supplementary Note 1). To determine the chemical composition of the as-grown crystals, energy dispersive X-ray spectroscopy (EDX) analyses were conducted on the surfaces of $Fe_{3-x}GaTe_2$ nanoflakes (Fig. 1a) that were exfoliated and placed onto the $Si_3N_4$ membrane (see Methods). The ratio of raw materials and the corresponding final crystal composition are listed in Table S1, Supplementary Fig. S1 and Fig. 1b. We found that the Fe deficiencies always exist in these crystals, while the minimum and maximum Fe contents correspond to $Fe_{2.84\pm0.05}GaTe_2$ and $Fe_{2.96\pm0.02}GaTe_2$, respectively. This result implies the feasibility of inducing Fe deficiency in the samples. To highlight the existence of Fe vacancies, the subsequent studies were focused on the minimum Fe content sample $Fe_{2.84\pm0.05}GaTe_2$. The Raman measurements revealed a clear red-shift of the $A_1$ peak in the non-stoichiometric $Fe_{2.84\pm0.05}GaTe_2$ to 126 $cm^{-1}$, in contrast to 130 $cm^{-1}$ observed for the stoichiometric $Fe_3GaTe_2$ reported previously (Fig. 1c)[35], which can be attributed to the localized effect of Fe-vacancy defects[38]. We conducted the magnetic characterization of the $Fe_{2.84\pm0.05}GaTe_2$ crystal with the external magnetic field (B) applied along the out-of-plane direction. Figure 1d displays the temperature-dependent magnetization (M-T) curve measured under a small magnetic field of 30 mT using a field-cooled protocol. By calculating the first derivative (d$M$/d$T$) of the M-T curve, we determined the Curie temperature ($T_c$) of the $Fe_{2.84\pm0.05}GaTe_2$ crystal to be ~ 350 K, which is slightly lower than that of the stoichiometric $Fe_3GaTe_2$. Additionally, an obvious magnetization kink was observed in the M-T curve at around 290 K, as indicated by the dashed box in Fig. 1d. This particular kink is typically regarded as a signature of rotation of the magnetic easy axis, and is often observed in skyrmion-hosting magnetic systems[23,39]. Moreover, the field-dependent magnetization curves for the $Fe_{2.84\pm0.05}GaTe_2$ sample with minimum Fe content reveal an out-of-plane easy magnetization direction at room-temperature (Fig. S2). These curves exhibit magnetic anisotropy almost identical to that of $Fe_{2.96\pm0.02}GaTe_2$ sample with high Fe content.

The crystal structure of the stoichiometric $Fe_3GaTe_2$ was confirmed to have the centrosymmetric space group of $P6_3/mmc$, and can be visualized as a series of Te-$Fe_3$Ga-Te monolayers stacked along the c-axis (Supplementary Fig. S3). Within each monolayer, the central $Fe_{ii}$-Ga slice flanked by two adjacent $Fe_i$ slices is sandwiched between two outer Te slices, indicating a protected $c \rightarrow -c$ mirror symmetry. Regarding the non-stoichiometric $Fe_{2.84\pm0.05}GaTe_2$ in this work, we analyzed the crystal structure on a microscopic scale (high-resolution scanning transmission electron microscopy, HR-STEM) and a macroscopic scale (single-crystal X-ray diffraction, XRD). Figure 1e presents a typical high-angle annular dark field (HAADF) image of the $Fe_{2.84\pm0.05}GaTe_2$ nanoflake along the [0001] zone axis. The image demonstrates that the $Fe_i$ atom columns (dark brown circles) are hexagonally surrounded by six Te-$Fe_{ii}$-Te-Ga atomic columns (marked with light brown and pale green balls), matching well with the standard $Fe_3GaTe_2$ crystal structure. Moreover, its lattice parameters ($a = b = 4.08$ Å) are slightly enlarged compared to those of the stoichiometric $Fe_3GaTe_2$ ($a = b = 3.99$ Å)[35]. These observations suggest that the Fe-deficiency does not cause noticeable lattice distortion in the ab plane. However, the HAADF image along the [11$\bar{2}$0] zone axis (Fig. 1f) reveals that $Fe_{ii}$ atoms deviate clearly from the center position of the Te slices along the c-axis, which is also supported by the annular bright-field (ABF-STEM) image in Fig. S4. By referencing the center of the two Te atoms in a magnified ABF-STEM image (Fig. S5), an averaged $Fe_{ii}$ deviation is calculated as $-0.16 \pm 0.06$ Å over an area of 2 × 17 unit cells (see Supplementary Note 2). Notably, this deviation can be also identified in the bottom layer of the $Fe_{2.84\pm0.05}GaTe_2$ unit cell, and both the offset and deviation direction are the same as those observed in the

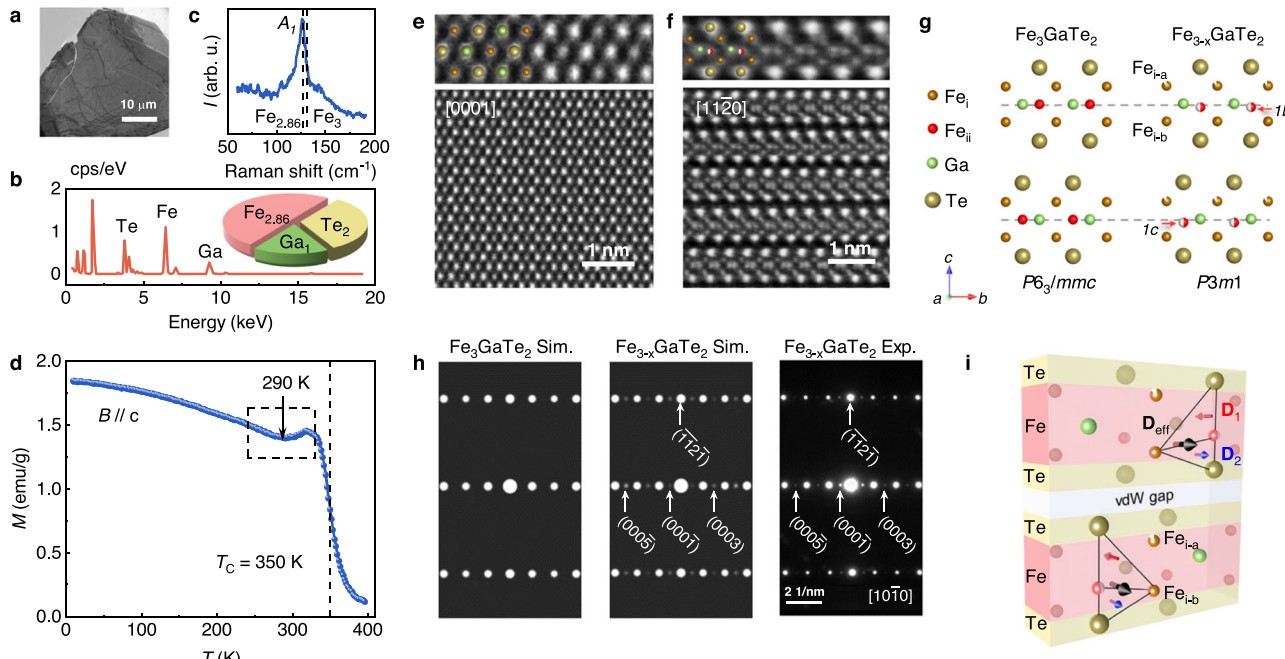

**Fig. 1 | Characterizations of Fe$_{2.84\pm0.05}$GaTe$_2$ single crystal. a** TEM image showing an exfoliated Fe$_{2.84\pm0.05}$GaTe$_2$ nanoflake on the Si$_3$N$_4$ membrane. **b** A typical EDX spectrum showing the chemical composition of the sample being Fe deficient. **c** Raman spectra for $A_1$ peak in Fe$_{2.84\pm0.05}$GaTe$_2$. The dashed line compares the $A_1$ peak shift with that of Fe$_3$GaTe$_2$[35]. **d** Temperature dependent magnetization curve $M(T)$ measured with field-cooled protocol at 10 mT. The dashed line marks the Curie temperature ($T_c = 350$ K). The magnetization kink highlighted using dashed box indicates a rotation of magnetic easy axis at around 290 K. **e** HAADF images viewed along the [0001] zone axis. The enlarged panel shows hexagonal arrangement of Fe$_i$ (dark brown), Te (light brown), Ga (pale green) without noticeable lattice distortion. **f** HAADF images along the [11$\bar{2}$0] zone axis. The enlarged panel shows an obvious displacement of Fe$_{ii}$ (red) atoms from Ga-Ga plane. **g** The comparison of crystal structure between Fe$_3$GaTe$_2$ and Fe$_{2.84\pm0.05}$GaTe$_2$ from side view. The red arrows at right panel indicate the Wyckoff site of deviated Fe$_{ii}$ atoms at 1c and 1b. **h** The comparison of simulated and experimental selected area electron diffractions (SAED) along [10$\bar{1}$0] zone axis. **i** Schematic illustration of DMI in asymmetric layers. The red arrow $D_1$ represents the direction of DMI vector in the upper triangle composed of Fe$_i$-Fe$_{ii}$-Te, while the blue arrow $D_2$ represents the lower part in the opposite direction. The black arrow $D_{eff}$ represents the sum of the non-zero DMI vector.

top layer. To gain additional insights into the structural features of the Fe$_{2.84\pm0.05}$GaTe$_2$, we performed single-crystal XRD on a single crystal with dimensions of approximately 3 × 3 mm². A total of 534 symmetry-independent reflections, corresponding to Miller indices of −5 ≤ $h$ ≤ 5, −5 ≤ $k$ ≤ 5, −20 ≤ $l$ ≤ 21, were collected for the structural determination and refinement. It is well known that for the centrosymmetric space group $P6_3/mmc$ of stoichiometric Fe$_3$GaTe$_2$, the reflection patterns, such as ($hh\bar{2}hl$) and (000$l$), are permitted only for even values of $l$ ($l = 2n$, where $n$ are integers), whereas they are forbidden for the odd values ($l = 2n + 1$)[33,40]. In the case of Fe$_{2.84\pm0.05}$GaTe$_2$, however, a series of weak (11$\bar{2}l$) and (000$l$) reflection patterns, such as (11$\bar{2}$3) and (11$\bar{2}$5) (see Fig. S6), were detected for $l = 2n + 1$, suggesting a substantial deviation from the original crystal structure of Fe$_3$GaTe$_2$. By fitting the reflections while allowing for the relaxation of $z$-positions and occupation ratio of Fe atoms, we obtained a symmetry-lowering structural model with the non-centrosymmetric space group $P3m1$ (Supplementary Note 2 and Fig. S7). The refined crystal structure revealed that the Fe$_{ii}$ atoms are deficient and located at the Wyckoff site 1c and 1b at ($x, y, z$) = (2/3, 1/3, 0.7482) and (1/3, 2/3, 0.2486) with a slight deviation of Fe$_{ii}$ atoms from the Ga-Ga plane, as shown in Fig. 1g. Based on the experimentally established crystal structure (Fig. S8a), we simulated the HR-STEM images along the [0001] and [11$\bar{2}$0] zone axis (Fig. S8b, c), which agree well with our experimental observations[41]. Moreover, we also simulated the selected area electron diffraction (SAED) along the [10$\bar{1}$0] and [11$\bar{2}$0] zone axes. The presence of odd $l$ values in the (000$l$) diffractions, evident in both simulated and experimental SAED results (Fig. 1h and Fig. S9), further confirms that the crystal structure of the Fe$_{2.84\pm0.05}$GaTe$_2$ belong to the non-centrosymmetric $P3m1$ space group, rather than the centrosymmetric $P6_3/mmc$ structure of Fe$_3$GaTe$_2$. These simulations agree well with our experimental

observations, providing solid evidential support for our structural model's reliability.

In comparison to the stoichiometric Fe$_3$GaTe$_2$ with a centrosymmetric structure, the presence of Fe deficiency in Fe$_{2.84\pm0.05}$GaTe$_2$ should exert a pivotal influence on the Fe$_{ii}$ deviation for the asymmetric structure. Our refined single-crystal XRD indicates that Fe deficiency is predominantly concentrated at the Fe$_{ii}$ positions with an occupancy ratio of 0.8467. Additionally, the upper-layer Fe$_{i-a}$ sites have an occupancy ratio of 0.9688, while the under-layer Fe$_{i-b}$ sites are nearly fully occupied (Fig. 1g). As observed in the line profile of Fe$_{i-a}$ and Fe$_{i-b}$ atoms in the ABF-STEM image (Fig. S5c), it is apparent that the image intensity of Fe$_{i-a}$ above Fe$_{ii}$ is weaker than that of Fe$_{i-b}$ below Fe$_{ii}$. Since the ABF imaging intensity is generally proportional to the number of projected atoms[42], the contrast difference between Fe$_{i-a}$ and Fe$_{i-b}$ indicates asymmetric site occupations, suggesting a small quantity of Fe vacancies in the Fe$_{i-a}$ site, which is consistent with the results of single-crystal XRD. To assess the influence of Fe$_{i-a}$ and Fe$_{ii}$ vacancies on Fe$_{ii}$ deviation, we further conducted first-principles calculations involving structure relaxation under three scenarios: no vacancy, Fe$_{i-a}$ vacancy, and Fe$_{ii}$ vacancy (Supplementary Note 3). The electron density of Fe$_{3-x}$Ga atoms is depicted in Fig. S10 to facilitate a comparison of the alterations in Fe$_{ii}$ chemical bonding: (a) The perfect Fe$_3$GaTe$_2$, with no Fe vacancy, showcases a hexagonally bonded Fe$_{ii}$-Ga plane. In this arrangement, the centrally positioned Fe$_{i-a}$ and Fe$_{i-b}$ dimers do not form direct bonds with Fe$_{ii}$ and Ga atoms. Thus, the overall chemical bonding is mirror-symmetric along the Fe$_{ii}$-Ga plane with no Fe$_{ii}$ deviation. (b) The presence of Fe$_{ii}$ vacancy induces deformation of the Ga electron density within the $ab$ plane. Nevertheless, no bonding is established between the Fe$_{ii}$-Ga plane and the Fe$_i$ atoms, which remain a mirror-symmetric electron density with no

Fe$_{ii}$ deviation. (c) In case of Fe$_{i-a}$ vacancy, there is additional electron-density overlapping between the lower Fe$_{i-b}$ atom and its three nearest Fe$_{ii}$ atoms, while no overlapping between Fe$_{i-b}$ and its three nearest Ga atoms. As a result, chemical bonding between the Fe$_{ii}$ and Fe$_{i-b}$ atoms induces a substantial Fe$_{ii}$ deviation, with a calculated $\delta c$ (Fe$_{ii}$ − Ga) of about −0.0554 Å, which compares favorably to the XRD result of −0.0871 Å. Furthermore, the calculated formation energy value for Fe$_i$ vacancy (2.96 eV/Fe) is higher than Fe$_{ii}$ vacancy (2.86 eV/Fe), indicating that the formation of Fe$_{ii}$ vacancies is more favorable. The above Fe vacancy model and calculated Fe$_{ii}$ deviation align well with the analysis from single-crystal XRD and ABF-STEM image. Therefore, we conclude that the asymmetric vacancy of Fe$_{i-a}$ induces a displacement of Fe$_{ii}$ atoms towards the −$c$ direction, which results in the symmetry breaking of the Fe$_{2.84\pm0.05}$GaTe$_2$ crystal structure.

Due to the asymmetric crystal structure, a pronounced DMI is naturally expected in the non-stoichiometric Fe$_{2.84\pm0.05}$GaTe$_2$. In quantitative terms, the DMI vector can be expressed as

$$\mathbf{D} = D \cdot \left( \hat{\mathbf{u}}_{ij} \times \hat{\mathbf{z}} \right), \tag{1}$$

where $D$ is the DMI constant, $\hat{\mathbf{u}}_{ij}$ represents the unit vector from Fe$_i$ atom to Fe$_{ii}$ atom, and $\hat{\mathbf{z}}$ represents the unit vector from magnetic Fe$_{ii}$ atom to heavy Te atom. For Fe$_{2.84\pm0.05}$GaTe$_2$, since its structural symmetry is broken by the Fe$_{ii}$ atom deviation, the DMI is proposed to originate from the interactions between the neighboring Fe$_i$-Fe$_{ii}$ pair and the adjacent Te atoms. As schematically shown in Fig. 1i, each Fe$_{2.84\pm0.05}$GaTe$_2$ unit cell has two distinct DMI sources: (a) the interaction between the Fe$_i$-Fe$_{ii}$ atom pair and the upper Te atom (corresponding to $\mathbf{D}_1$ vector, represented by the red arrow) and (b) the interaction between the Fe$_i$-Fe$_{ii}$ pair and the lower Te atom (corresponding to $\mathbf{D}_2$ vector, represented by the blue arrow). Equation (1) indicates that the two DMI vectors $\mathbf{D}_1$ and $\mathbf{D}_2$ are perpendicular to the Fe$_i$-Fe$_{ii}$-Te triangle of atoms. This is consistent with the Fert-Levy DMI observed at heavy metal/ferromagnet interfaces, which can lead to the formation of Néel-type skyrmions[2]. Moreover, we found that the directions of $\mathbf{D}_1$ and $\mathbf{D}_2$ are opposite due to the opposite directions of $\hat{\mathbf{z}}$ for $\mathbf{D}_1$ and $\mathbf{D}_2$. For the centrosymmetric Fe$_3$GaTe$_2$, because the Fe$_{ii}$ atom is located at the center of two adjacent Te atoms, the upper $\mathbf{D}_1$ and lower $\mathbf{D}_2$ vectors would always cancel out with each other, resulting in an effective net DMI vector ($\mathbf{D}_{eff}$) of zero. In the case of Fe$_{2.84\pm0.05}$GaTe$_2$, however, the deviation of Fe$_{ii}$ atoms shows −0.0871 Å atom displacement towards −$c$ directions, which breaks the inversion symmetry, and thus makes the nonequal $\mathbf{D}_1$ and $\mathbf{D}_2$ yield a nonzero $\mathbf{D}_{eff}$ within each monolayer. As for the total DMI ($D_{total}$) of the unit cell, since the $\mathbf{D}_{eff}$ vectors in top and bottom layers have the same direction, the magnitude of $D_{total}$ is the sum of the two vectors. Moreover, Fig. S11b illustrates the effective net DMI vectors $D_{eff}$ viewed from [0001] zone axis, which are perpendicular to the Fe$_i$-Fe$_{ii}$-Te atom cross sections and exhibit threefold rotational symmetry within the $ab$ plane. Based on the model depicted in the aforementioned illustration, we quantitatively investigated the relationship between the Fe$_{ii}$ deviation value $\delta c$(Fe$_{ii}$ − Ga) and the DMI constant $D$ (see Supplementary Note 4 and Fig. S12 for details). For a centrosymmetric structure ($\delta c = 0$), the absence of Fe$_{ii}$ deviation yields $D = 0$ mJ/m$^2$. Conversely, a non-centrosymmetric structure ($\delta c = -0.0871$ Å), determined by single-crystal XRD, corresponds to $D = 0.91$ mJ/m$^2$. This significant DMI constant meets the crucial requirement for the formation of skyrmions[43–46], suggesting that the non-stoichiometric Fe$_{2.84\pm0.05}$GaTe$_2$ has the potential to exhibit topological magnetism.

For a magnetic system hosting topological spin configurations, the total Hall resistivity ($\rho_{xy}$) typically comprises three components[4,47]:

$$\rho_{xy}(H) = \rho_{xy}^N + \rho_{xy}^A + \rho_{xy}^T = R_0H + S_A\rho_{xx}^2M(H) + \rho_{xy}^T, \tag{2}$$

where $\rho_{xy}^N = R_0H$ is the normal Hall resistivity, $R_0$ is the normal Hall coefficient; $\rho_{xy}^A = S_A\rho_{xx}^2M(H)$ is the anomalous Hall resistivity, $S_A$ is the scaling coefficient, $\rho_{xx}$ is the longitudinal resistivity, $M(H)$ is the magnetic field-dependent magnetization; and $\rho_{xy}^T$ is the topological Hall resistivity. Of these components, $\rho_{xy}^T$, which is driven by the local spin chirality, is widely regarded as a key transport signature of topological spin configurations. To investigate the potential existence of $\rho_{xy}^T$ in Fe$_{2.84\pm0.05}$GaTe$_2$, we fabricated a Hall device with a sample thickness of 250 nm (Fig. 2a) and conducted measurements of $\rho_{xy}$ with the external magnetic field applied along the normal direction of the device over the temperature range of 350 − 10 K, as shown in Fig. 2b. The results indicate that $\rho_{xy}$ varies in a nonlinear manner with sweeping the magnetic field, suggesting the presence of a ferromagnetic-order induced anomalous Hall effect. We subsequently fitted $\rho_{xy}$ using the $M(H)$ and $\rho_{xx}(H)$ curves in the high-field region based on Eq. (2). Figure 2c displays both the fitting and experimental $\rho_{xy}(H)$ curves at 300 K. It is clear that there is a discrepancy at the low-field region, indicating the presence of a pronounced THE component in $\rho_{xy}$. The extracted magnetic field-dependent $\rho_{xy}^T$ curves over the temperature range of 350 − 10 K are summarized in Fig. 2d. It is important to note that the maximum value of $\rho_{xy}^T$ (-1.13 μΩ cm) at 300 K is one order of magnitude higher than that of the material systems hosting RT skyrmions, such as Co-doped 2D magnets (Fe$_{0.5}$Co$_{0.5}$)$_5$GeTe$_2$ ($\sim 0.8$ μΩcm)[31], Kagome ferromagnets Fe$_3$Sn$_2$ (- −0.4 μΩ cm)[47], and Ir/Fe/Co/Pt multilayers (-0.03 μΩ cm)[48], and is comparable to that of noncoplanar ferromagnet Cr$_5$Te$_6$ (-1.6 μΩ cm) at 90 K[49]. Furthermore, we investigated thickness- and magnetic field-dependent Hall resistivity $\rho_{xy}$ and topological Hall resistivity $\rho_{xy}^T$ under room temperature (Fig. S13). As the sample thickness increases, the topological Hall signals gradually strengthen and shift towards higher magnetic field. More importantly, the THE signals persist over a broad temperature range and various thickness, suggesting the existence of topological spin configurations in Fe$_{2.84\pm0.05}$GaTe$_2$.

To directly visualize the possible topological spin textures associated with the topological Hall effect, we conducted cryo-LTEM experiments on an exfoliated [0001]-oriented Fe$_{2.84\pm0.05}$GaTe$_2$ nanoflake with a thickness of approximately 100 nm, as schematically illustrated in Fig. 3a. For the cryo-LTEM measurements, we heated the sample above $T_c$ (380 K) and then cooled it to the desired temperature with a liquid nitrogen TEM holder under a 30 mT out-of-plane magnetic field. Following this field-cooling procedure, no magnetic contrast in the Lorentz phase images was observed when the electron beam was injected along the normal direction of the nanoflake at 300 K (Fig. 3b and Supplementary Fig. S14). However, if we tilted the nanoflake an angle ($\theta$) of ±20° away from the horizontal plane, bubble-like domains with half-dark/bright contrast were detected. It is well known that the magnetic contrast discernible in a Lorentz phase image acquired via Fresnel imaging mode is predominantly caused by the deflection of the electron beam due to the in-plane magnetic field as it passes through a magnetic sample. Since in the Bloch-type domain walls the magnetization, or spin, direction is out-of-plane, the Lorentz deflection is thus in-plane and perpendicular to the walls, yielding clear domain walls contrast. In contrast, in the Néel-type domain walls, the magnetization direction in the walls is in-plane and the Lorentz deflection is along the wall length, resulting in no contrast. However, once the sample is tilted, this is no longer the case as an out-of-plane magnetization component is generated[50–52]. The angle-dependent contrast modulation seen in our experiments unambiguously suggests that the bubble-like domains we imaged are Néel-type. We further analyzed their in-plane spin structures using the transport-of-intensity equation (TIE)[53], as displayed in the right panel of Fig. 3b. We found that the in-plane magnetic induction was composed of a pair of conjoined clockwise and counterclockwise spin swirls, which agrees well with the calculated magnetic induction map for the Néel-type skyrmions[31,52]. On the basis of the deduced double in-plane swirls and

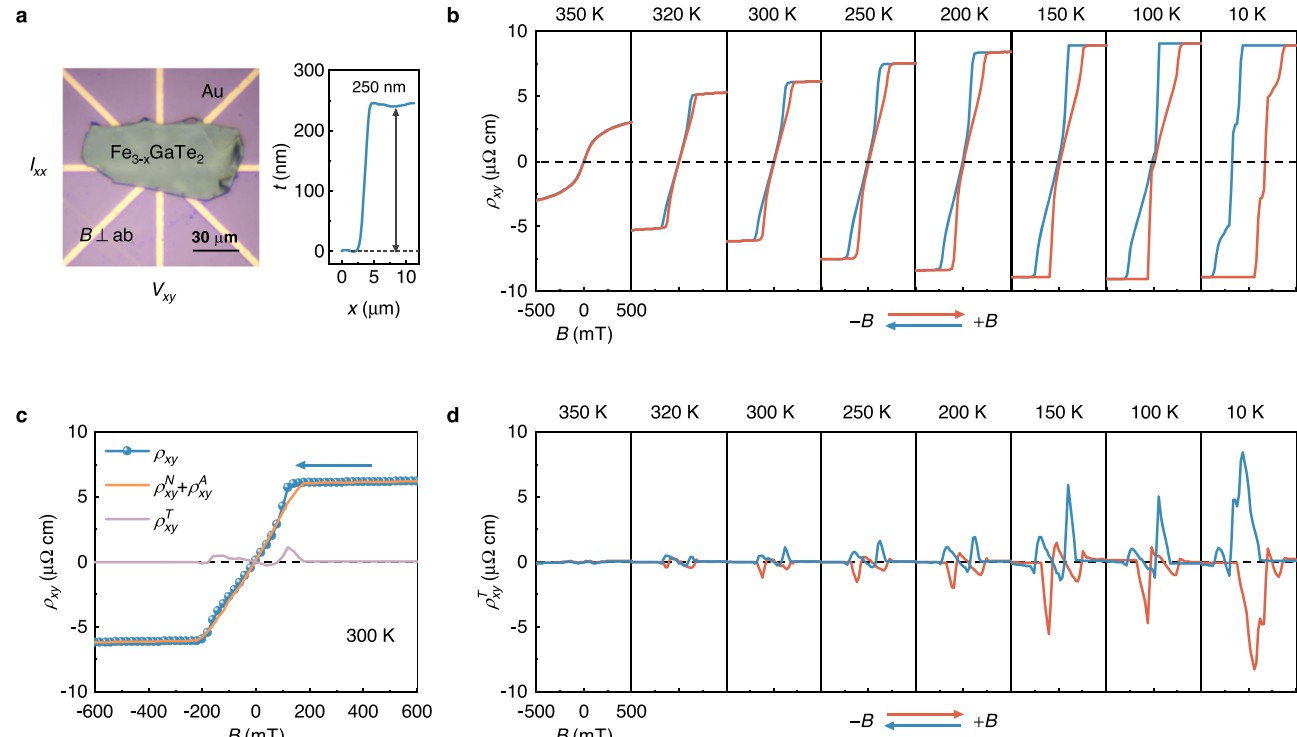

**Fig. 2 | Transport properties of Fe$_{2.84\pm0.05}$GaTe$_2$ nanoflake. a** Optical image of Fe$_{2.84\pm0.05}$GaTe$_2$ Hall device with a sample thickness of 250 nm. A current $I_{xx}$ was applied across the sample plane and transverse voltages $V_{xy}$ were measured simultaneously. **b** Magnetic hysteresis of Hall resistivity $\rho_{xy}$ at various temperatures from 350 to 10 K. Red (blue) curves were measured with increasing (decreasing) magnetic field. **c** Extraction procedure of normal Hall resistivity $\rho_{xy}^N$, anomalous Hall resistivity $\rho_{xy}^A$ and topological Hall resistivity $\rho_{xy}^T$ at 300 K. The blue arrow indicates the sweep direction of the magnetic field pointing downward. **d** Magnetic hysteresis of topological Hall resistivity $\rho_{xy}^T$ as a function of magnetic field at various temperatures from 350 K to 10 K. Red (blue) curves were extracted with increasing (decreasing) magnetic field.

Néel-type spin arrangements, we conclude that the observed bubble-like domains are indeed Néel-type skyrmions.

Due to the high $T_c$ of Fe$_{2.84\pm0.05}$GaTe$_2$, its magnetic phase is expected to remain stable well above room temperature. The upper panel of Fig. 3c presents the Lorentz phase images taken at a zero magnetic field over the temperature range of 300 to 340 K after the field-cooling operation (see Methods). The results indicate that the skyrmion phase remain stable up to 330 K, which represents a record-high value compared with that of the skyrmion-hosting magnetic vdW materials reported to date. However, when the temperature exceeds 330 K, the magnetic skyrmions become elongated and gradually transformed into stripe domains due to the presence of thermal fluctuations, which are proposed to be large enough to overcome the topologically protected energy barrier between skyrmions and stripe domains above 330 K. Furthermore, the magnetic field-dependent domain evolution process after the field-cooling operation was studied within a temperature range of 340–100 K (lower panel of Fig. 3c and Supplementary Fig. S15). To illustrate the correlation between the magnetic states and both the magnetic field and temperature, we construct a magnetic phase diagram (Fig. 3d), in which the red region denotes that the sample hosts high-density skyrmions at the corresponding $T$-$B$ plane, while the blue region indicates the absence of skyrmions. As displayed in Fig. 3d, high-density, field-free skyrmions can be stabilized in a wide temperature range of 100–330 K, demonstrating that Fe$_{2.84\pm0.05}$GaTe$_2$ is a promising material platform for use in spintronic devices.

We further use magnetic force microscopy (MFM), which is sensitive to the out-of-plane magnetic field in the sample, to study the effect of nanoflake thickness on the stabilization of the skyrmion phase at room temperature after field-cooling manipulation. All the nanoflakes used for the MFM measurements were freshly exfoliated from

the same batch to ensure consistency (Supplementary Fig. S16). Additionally, we used a low-moment MFM tip (<10 mT) to minimize the influence of the tip's magnetic field on the domain structures during scanning. Figure 4a illustrates a series of MFM images that record the magnetic field-dependent domain evolution processes at six typical thicknesses. Notably, the vertically arranged MFM images represent the same sample area under different out-of-plane fields, while the horizontal direction represents the variation in thickness for different samples. For the 250 nm thick sample, the skyrmions are arranged densely into honeycomb lattices with a large skyrmion size ($d_{sk}$) of approximately 180 nm at zero magnetic field. As the external magnetic field increases, there is little change in $d_{sk}$, accompanied, however, by a decrease in density with the sudden annihilation of skyrmions at 210 mT. When the sample reaches saturation magnetization, the MFM images with uniform contrast represent the fully ferromagnetic (FM) state. Varying the sample thickness could significantly affect the skyrmion size due to the change in the strength of dipole-dipole interaction[33,54]. Upon decreasing the sample thickness, the skyrmion size at a zero magnetic field reduces correspondingly, reaching a minimum skyrmion size of 87 nm when the sample thickness is decreased below 100 nm (Fig. 4b). This value is much smaller than that of the skyrmions in other skyrmion-hosting vdW magnets reported to date, such as Cr$_{1+x}$Te$_2$ (~400 nm)[34], Fe$_3$GeTe$_2$ (~250 nm)[55], Fe$_5$GeTe$_2$ (~200 nm)[24], and (Fe$_{0.5}$Co$_{0.5}$)$_5$GeTe$_2$ (~150 nm)[31], as displayed in Fig. 4c (see Supplementary Fig. S17 for detailed size distribution of field-free skyrmions in Fe$_{2.84\pm0.05}$GaTe$_2$ at varied temperatures, and Fig. S18 for the corresponding Micromagnetic simulations). Figure 4d provides a comprehensive overview of the RT skyrmion phase diagram, depicting the relationship between thickness, magnetic field, and the occurrence of skyrmion states. The field-free skyrmions are stable in a broad thickness range and the highest density appears to be between 46 and

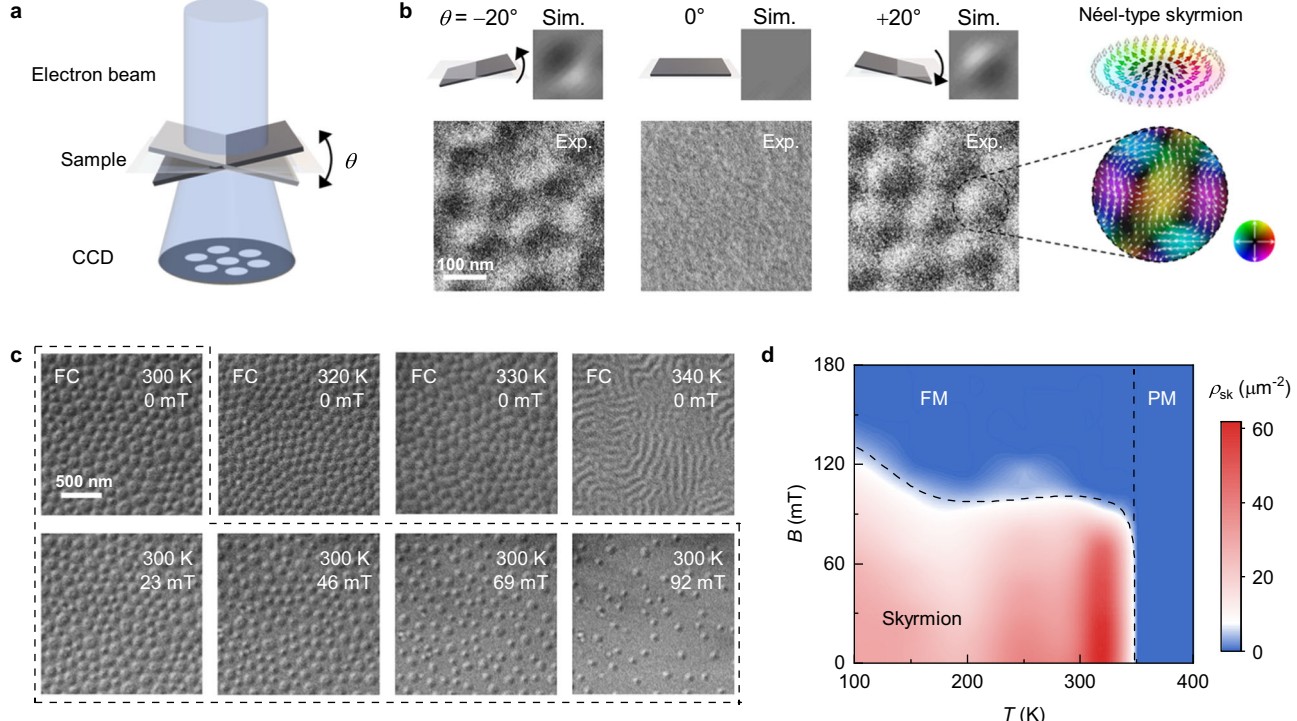

**Fig. 3 | LTEM measurements of Néel-type skyrmions in Fe$_{2.84\pm0.05}$GaTe$_2$.**
**a** Schematic diagram of LTEM indicating the tilt angle $\theta$ with respect to the sample plane. All images were acquired with a defocus value of $d = 2$ mm. **b** Experimental and simulated Lorentz phase images of RT Néel-type skyrmions at ±20° and 0° tilt. The sample was previously applied field cooling (FC) with $B = 30$ mT. The illustration at the top utmost right panel shows a typical spin configuration of Néel-type skyrmions, while the bottom utmost right panel shows the corresponding magnetic

induction field map at +20° tilt. **c** Temperature and magnetic field dependence of the skyrmion state. Note that the top row of Lorentz phase images at various temperatures were acquired individually after FC. The Lorentz phase images bounded by the dashed line represent the RT skyrmions evolution with increasing magnetic field. **d** Skyrmion phase diagram. The color indicates the skyrmion density $\rho_{sk}$. The black dashed line shows the boundary between skyrmion and ferromagnetic phase.

60 nm. As the thickness of Fe$_{2.84\pm0.05}$GaTe$_2$ nanoflake decreases to 25 nm, the stripe domains remain while the field-cooling process no longer generates skyrmions. Compared with the previously reported skyrmion-hosting 2D material (Fe$_{0.5}$Co$_{0.5}$)$_5$GeTe$_2$[31], Fe$_{2.84\pm0.05}$GaTe$_2$ exhibits smaller magnetic parameters such as DMI constant $D$, saturation magnetization $Ms$, threshold of sample thickness $t$ and etc. (see Fig. S19 and Supplementary Note 5 for the determination of magnetic parameters), which contribute to the reduction in skyrmion size (see Table S3, Fig. S20 and Supplementary Note 6 for the corresponding micromagnetic simulations). Moreover, our micromagnetic simulations demonstrate the influence of in-plane magnetic field $B$ on skyrmion shape, revealing a progressive transformation from a circular to an elliptical configuration (see Fig. S21 and Supplementary Note 7).

Achieving ultrafast and energy-efficient writing of skyrmions is a critical requirement in the pursuit of their practical applications in high-speed and low-power spintronic devices. Conventional field-cooling operations are unsuitable as they involve heating the entire sample above the $T_c$ and slow cooling to the desired temperature[55–57]. In contrast, fs laser pulses can demagnetize a localized region on the micrometer scale within fs timescale followed by a picosecond (ps) thermal quenching[58–64]. This unique approach enables the creation of metastable magnetic states at extremely short timescales, offering a promising pathway for the ultrafast writing of skyrmions. Several studies have reported successful writing of skyrmions using short laser pulses in various materials, including magnetic multilayer films[60–62], FeGe[63], and Co$_9$Zn$_9$Mn$_2$[64]. To further investigate the possibility of ultrafast writing of RT skyrmions in the 2D van der Waals Fe$_{2.84\pm0.05}$GaTe$_2$ nanoflakes, we performed Lorentz phase microscopy measurements under fs laser excitation using our homemade in-situ optical LTEM setup that enables single-shot fs laser pulse excitation on

the sample (520 nm wavelength, 50 μm focal spot size, and 300 fs pulse duration), as illustrated in Fig. 5a. We recorded the fs laser writing processes by taking snapshots of Lorentz phase images before and after the laser pulse excitation (fluence of ∼ 11 mJ/cm²), while the electron beam was blanked during laser pulse excitation to avoid unexpected sample damage. Upon laser pulse excitation of the initial domains, ultrafast heating with electron temperature ($T_{elec}$) above $T_c$ would temporarily melt the long-range magnetic order (Fig. 5b) on the fs time scale[59,60,63]. Subsequently, the paramagnetic state is rapidly quenched on the ps timescale due to the electron-phonon coupling, where the phonon temperature ($T_{phon}$) follows the variation of $T_{elec}$ (see Supplementary Note 8 and Fig. S22 for detailed calculations of temperature evolutions based on a two-temperature model). Such quenching process leads to a rapid decrease of the temperature to below $T_c$ and initiates the remagnetization process of the spin system. Since only skyrmion is a stable solution for the Fert-Levy DMI observed in Fe$_{2.84\pm0.05}$GaTe$_2$, it is expected that skyrmion states could be created during the thermal relaxation in the quenching process.

To demonstrate the feasibility of using a fs laser pulse to write RT skyrmions in the 2D van der Waals Fe$_{2.84\pm0.05}$GaTe$_2$ material, we conducted a step-by-step single-shot fs laser pulse excitation measurement on different initial field-polarized magnetic states, including stripe and single domains. These initial field-polarized domain structures were achieved by varying the out-of-plane magnetic field from a saturation state (±2 T), as indicated by the normalized Hall hysteresis curve in Fig. 5c. Moreover, the corresponding Lorentz phase images captured the magnetic states before and after the laser pulse excitation at six representative magnetic fields (see Supplementary Figs. S23, S24 for more details). These images provide compelling evidence that the presence of an external assisting magnetic field is vital for the

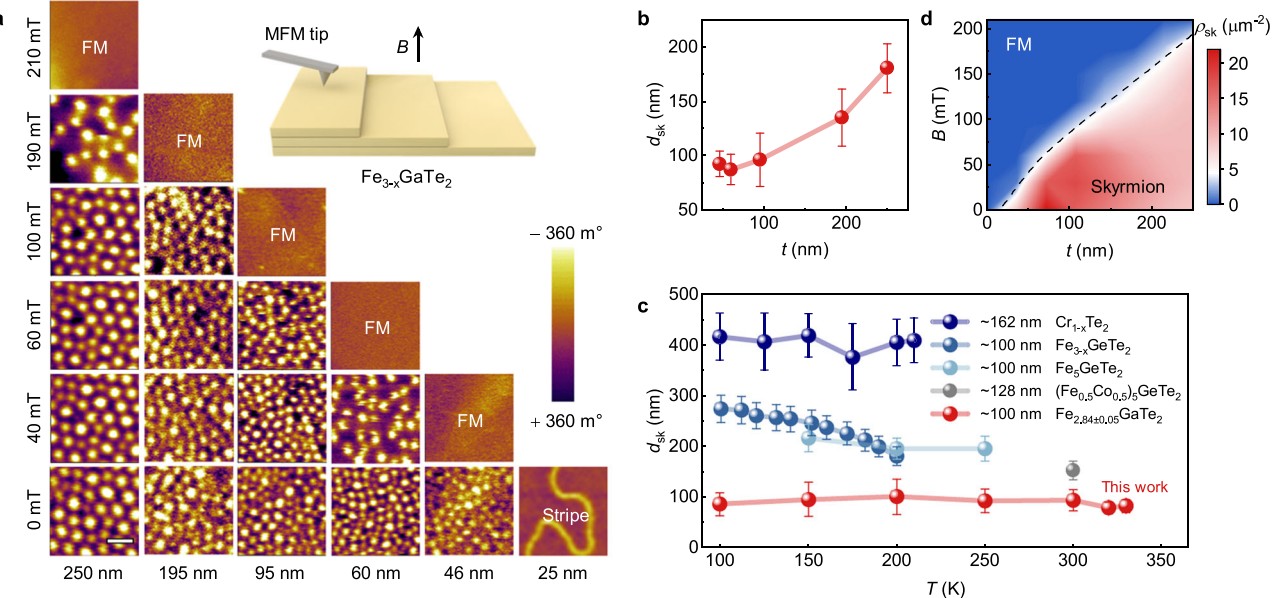

**Fig. 4 | Thickness-tunable skyrmions imaged by MFM. a** Typical MFM images of RT skyrmions taken at different thicknesses and external magnetic fields. In these images, bright contrast in the images corresponds to spin down, while dark contrast corresponds to spin up relative to the sample normal direction. The inset shows the schematic for the MFM experiment with magnetic field perpendicular to the sample plane. **b** RT skyrmion diameter $d_{sk}$ versus sample thickness $t$ at zero field. **c** Comparison for diameter $d_{sk}$ of field-free skyrmions and temperature $T$ for various 2D vdW materials with a thickness around 100 nm, including $Cr_{1+x}Te_2$[34], $Fe_{3-x}GeTe_2$[55], $Fe_5GeTe_2$[24], $(Fe_{0.5}Co_{0.5})_5GeTe_2$[31], and $Fe_{2.84\pm0.05}GaTe_2$ in this work. Error bars represent the standard error of skyrmion sizes averaged at various temperature. **d** Thickness and magnetic field dependence of RT skyrmion phase diagram. The color indicates the skyrmion density $\rho_{sk}$. The black dashed line shows the boundary between skyrmion and ferromagnetic phase.

successful fs laser writing of skyrmions in the material[60–63]. At $B = 0$ mT, only stripe domains were observed after the laser excitation (fluence of ~11 mJ/cm²). As $B$ is swept towards positive saturation, the width of the stripe domains gradually decreases. After the fs laser excitation, the magnetic states accessible by the laser pulse initially exhibit a mixed state of stripe domains and skyrmions at 11 mT and then transition to fully-formed skyrmions with the highest density at 46 mT. As $B$ is increased to 80 mT, the initial stripe domains start to fade out, and the laser-accessible skyrmions completely vanish, transforming into a single-domain state. Compared to the single domain state stabilized by a pure magnetic field ($B = 104$ mT), the assisting magnetic field decreases significantly, suggesting that for the laser-writing operation the laser functions as an efficient field that can effectively shift the magnetic states, possessing a higher energy state in the magnetization process to a lower energy state. When the magnetic field is increased to 104 mT, the initial stripe domains transform into a single domain state, and the laser pulse with the used fluence can no longer induce a magnetic phase transition. However, as the magnetic field is decreased from the positive saturation field, the single domain state persists at a much lower magnetic field of 46 mT due to the hysteresis effect. Interestingly, we find that there exists a magnetic field window (69 mT to 46 mT) where laser-accessible skyrmions can be created from the initial single domain state, even though it is in a higher energy state than that of the skyrmions in the magnetization process. As the magnetic field continues to decrease, the behavior under laser excitation in the low-field region (35 mT to 0 mT) is similar to that at the beginning of the positive field sweep (0 mT to 35 mT). These results indicate that the laser-writing operation is independent of the initial domain structures but rather dependent on the strength of the assisting magnetic field. Furthermore, conventional zero-field cooling can only result in the formation of interconnected, relatively long stripe domains, but does not spontaneously lead to the creation of skyrmions[57]. To demonstrate the differences with in-site fs laser quenching approach, we conducted fluence-dependent laser pulse excitation without magnetic field (see Fig. S25 and Supplementary

Note 9). After a single laser pulse with fluence of 1.3 mJ/cm², stripe domains show slightly domain wall movement. Upon increasing to 9.4 mJ/cm², a hybrid state with both stripes and skyrmions are formed, while at 11 mJ/cm² only stripe domains observed. This indicates that a higher fluence of fs laser can completely demagnetize the sample during the laser-writing process, regardless of the initial magnetic state.

To further understand the underlying magnetization dynamic process of the ultrafast laser writing of skyrmions in $Fe_{2.84\pm0.05}GaTe_2$ under external magnetic field assistance, we performed finite-element micromagnetic simulations on the subsequent magnetic structure evolution of an initial stripe domain after a fs laser pulse excitation at a certain external magnetic field based on the Landau-Lifshitz-Gilbert equation with Langevin dynamics[65–67]. We considered the following scenarios for the simulations: (i) the fs laser pulse interacts with the magnetic structures through photothermal effect; and (ii) the fs pulse heats the sample above the $T_c$ and melts the electronic spin structures, but without changing the atomic lattice. Specifically, the laser quenching-induced magnetization dynamics was achieved by relaxing the magnetic system from the laser-induced paramagnetic state under an out-of-plane magnetic field of 46 mT (see Supplementary Note 5 and Fig. S19 for more details about the simulations). As shown in Fig. 5d (see details in Supplementary Movie 1), following the excitation by the femtosecond (fs) laser pulse, the initial melted spin state (snapshot at $t_0$) rapidly evolves into numerous nanoscale spin clusters. These clusters contain topological defects, including skyrmionic and anti-skyrmionic nucleation centers (snapshot at $t_1$). This transformation occurs due to the ultrafast cooling, achieved at a quenching rate of up to $10^{12}$ K/s[58,68]. Because only skyrmion is a stable solution for the Fert-Levy DMI observed in $Fe_{2.84\pm0.05}GaTe_2$, in the further cooling process the anti-skyrmionic nucleation centers merge and annihilate with the nearby skyrmionic nucleation centers and appear less frequently until completely disappear, namely, the topological fluctuations[60,69], resulting in formation of a pure skyrmionic state (snapshot at $t_2$). The

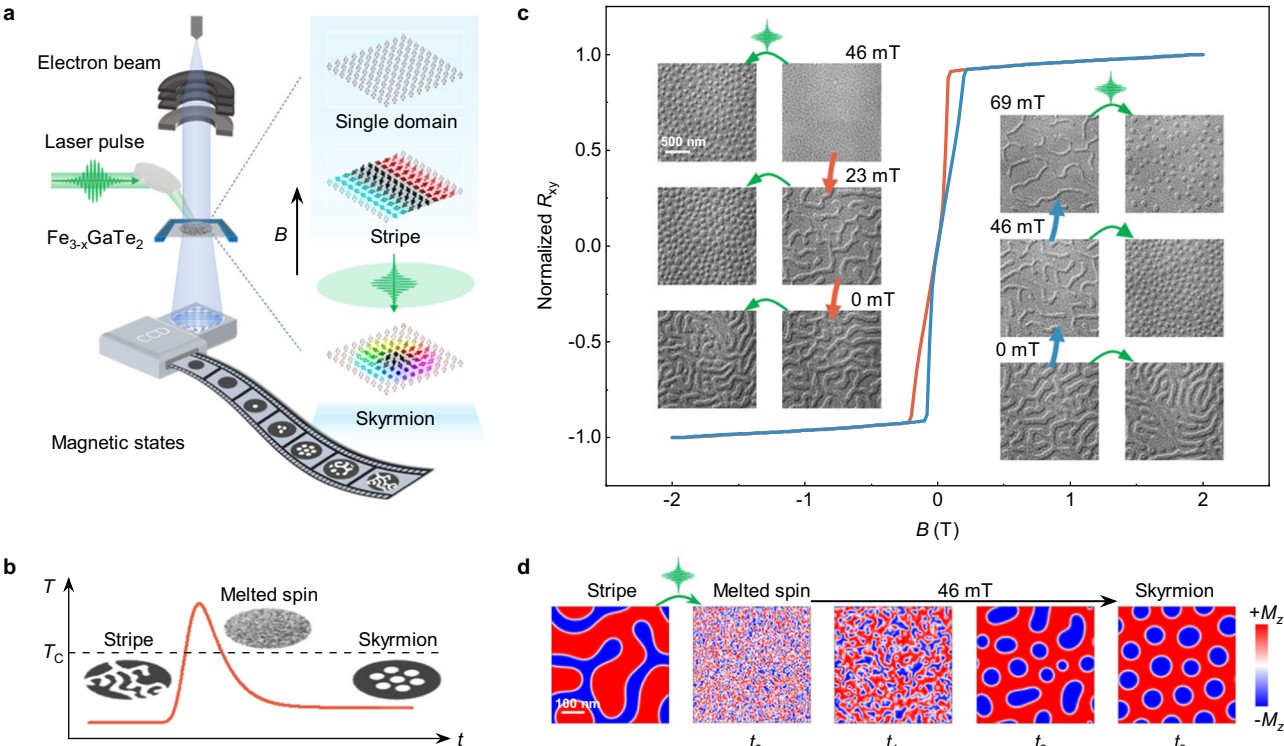

**Fig. 5 | Ultrafast fs laser writing of skyrmions in Fe$_{2.84\pm0.05}$GaTe$_2$. a** Schematic of the in-situ optical LTEM experiments. The sample was exposed to a single-shot fs laser pulse with an applied out-of-plane magnetic field. The defocused Lorentz phase images ($d$ = 2 mm) captured the magnetic state at each step before and after the fs-laser pulse excitation. The right panel shows the typical spin configurations of stripe, single domain, and Néel-type skyrmions. **b** Schematic of the ultrafast demagnetization process for writing skyrmions from an initial stripe domain by a fs laser pulse. The sample is temporarily laser-heated above $T_c$ to melt the existing spin ordering, followed by quenching with external magnetic field to form a new spin ordering, skyrmions. **c** Illustration of single-shot laser pulse excitation from field-polarized magnetic state to subsequent laser-accessible magnetic states. The blue and red lines indicate the magnetization states during field swapping of Hall hysteresis curve. The inset shows corresponding Lorentz phase images before and after fs-laser pulses. **d** Simulated laser-induced stripe domain to skyrmion evolution process under an out-of-plane magnetic field of 46 mT. The magnetization along the z-axis ($M_z$) is represented by the color bar (+$M_z$ in red and −$M_z$ in blue).

skyrmion nucleation process is complete at $t_2$, but the skyrmion size has not reached the energy equilibrium state. In order to minimize the magnetostatic energy, the domain walls move and gradually evolve into a uniform skyrmion lattice (snapshot at $t_3$). The micromagnetic simulations reproduce well the magnetic phase transitions observed in our experiments, confirming the reliability of our established relationship between the laser-writing process, magnetic field strength, and magnetic domain evolution. Such evidenced controllable ultrafast laser writing of skyrmions in 2D van der Waals magnetic materials provides opportunities for both fundamental researches and device applications towards magneto-optical control of spin topologies.

We report the discovery of a field-free sub-100 nm Néel-type skyrmion state in non-stoichiometric Fe$_{2.84\pm0.05}$GaTe$_2$ over a broad temperature range from 330 K to 100 K. Using HR-STEM and single-crystal XRD, we determine that the deviation of Fe$_{ii}$ atoms from the center position of the Te slices due to the asymmetric Fe$_{i-a}$ vacancies induces a transformation of the crystal structure from centrosymmetric to non-centrosymmetric, enabling the formation of skyrmions through the in-plane isotropic DMI. LTEM (along with MFM) shows the size of the skyrmions decreases with sample thickness, and a field-free sub-100 nm skyrmion state was achieved at RT within a specific sample thickness range of 40 nm to 60 nm. Furthermore, we demonstrate that a single fs-laser pulse can rapidly generate field-free sub-100 nm skyrmions from both stripe domains and single domains. Our study demonstrates not only the non-stoichiometric Fe$_{2.84\pm0.05}$GaTe$_2$ to be a promising material platform for exploring magnetic skyrmions, but also the fs-laser can be a powerful tool to manipulate and control

topological chiral spin textures to realize skyrmion-based high-speed logic and memory applications.

## Methods
### Single crystal growth and structure characterization
Single crystals of Fe$_{3-x}$GaTe$_2$ were grown by the self-flux method. The mixtures of Fe (99.99%), Ga (99.99%), and Te (99.99%) elements were mounted in an alumina crucible and sealed inside a quartz tube under high vacuum (-10$^{-4}$ Pa). The mixtures were firstly heated at 1150 °C for 24 h, then followed by slow-cooling down to 850°C for 3 weeks. Finally, excessive molten flux was centrifuged to separate the single crystals. Its chemical composition was determined by energy dispersive x-ray spectroscopy mapping (EDX, Bruker Nano GmbH Berlin). A comprehensive overview of the raw material composition and the final crystal composition are listed in Supplementary Note 1 and Table S1. The single-crystal X-ray diffraction was carried out with a four-circle diffractometer (Bruker D8 venture). The refined crystal structure was solved and refined by using the Bruker SHELXTL Software Package. The HR-STEM images were acquired by high-resolution transmission electron microscope (HRTEM, JEOL ARM200F). Magnetometry measurements were carried out with the Quantum Design PPMS.

### LTEM and MFM measurements
The fresh Fe$_{2.84\pm0.05}$GaTe$_2$ nanoflakes utilized for Hall devices, LTEM and MFM experiments were prepared through an all-dry mechanical-transfer method within an argon-filled glovebox. In an argon-protected environment, the nanoflakes were first produced on PDMS stamp by micromechanical cleavage, and then transferred

onto $Si_3N_4$ membrane or $SiO_2$/Si substrate, with or without pre-patterned Au electrodes. To maintain the integrity of the samples, these freshly prepared Hall devices, LTEM, and MFM specimens were promptly placed into a plastic box within the argon-filled glovebox and securely sealed with parafilm before removal. During subsequent measurements outside the glovebox, the exposure of the samples to air was minimized, and all the sample transfers were conducted within a confined timeframe of no more than 10 minutes. LTEM measurements were performed on Thermo Fisher Talos F200i at an acceleration voltage of 200 kV. The objective lens was used to apply a magnetic field perpendicular to the sample plane by controlling the excitation current. The specimen was in-suit warmed and cooled by a liquid nitrogen double-tilt sample holder. In order to perform field cooling with accurate temperature control, the sample was initially cooled to +20 K above the desired temperature at a rate of −5 K/min, and then slowly cooled down to the desired temperature at a rate of −1 K/min. The in-plane magnetization distribution map was reconstructed from the under- and over-focused images using the transport-of-intensity equation (TIE) approach. MFM measurements were performed by scanning probe microscopy (MFP-3D, Asylum Research), which equipped a low-moment magnetic tip (PPP-LM-MFMR, Nanosensors) and VFM3 component (Asylum Research). To protect samples from air degradation, Hall and LTEM measurements were conducted under vacuum conditions, while MFM measurements were carried out in an environment continuously flushed with argon gas to ensure effective protection.

### In-situ optical LTEM experiments

To directly visualize the fs laser writing of RT skyrmions in the 2D van der Waals $Fe_{2.84\pm0.05}GaTe_2$ nanoflakes, we performed the in-situ optical LTEM experiments in our homemade 4D-electron microscopy (Thermo Fisher Talos F200i), which enables single-shot fs laser pulse excitation on the sample under the Lorentz phase imaging mode. The Lorentz phase images were acquired under the Fresnel mode, in which the external perpendicular magnetic field was applied by the objective lens with controlled lens current. The fs laser system was triggered externally with a digital delay generator which outputted single-shot fs laser pulses with 520 nm wavelength, 300 fs pulse duration, and 11 mJ/$cm^2$ fluence, where the laser spot size was adjusted to be 50 μm to ensure homogeneous illumination on the sample.

### Reporting summary

Further information on research design is available in the Nature Portfolio Reporting Summary linked to this article.

## Data availability

The single crystal X-ray diffraction, Lorentz transmission electron microscopy, and characterization data generated in this study are provided in the Supplementary Information file. The data that support the findings of this study are available from the corresponding authors upon reasonable request.

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

## Acknowledgements

This work was supported by the National Key Research and Development Program of China at grant No. 2020YFA0309300, Science and Technology Projects in Guangzhou (Grant No. 202201000008), the National Natural Science Foundation of China (NSFC) at grant No. 12304146, 11974191, 12127803, 52322108, 52271178, U22A20117, and 12241403, China Postdoctoral Science Foundation (2023M741828), Guangdong Basic and Applied Basic Research Foundation (Grant nos. 2021B1515120047 and 2023B1515020112), the Natural Science Foundation of Tianjin at grant No. 20JCJQJC00210, the 111 Project at grant No. B23045, and the "Fundamental Research Funds for the Central Universities", Nankai University (Grant Nos. 63213040, C029211101, C02922101, ZB22000104, and DK2300010207). This work was supported by the Synergetic Extreme Condition User Facility (SECUF). The work at Brookhaven National Laboratory was supported by the Materials Science and Engineering Divisions, U.S. DOE-BES under Contract No. DE-SC0012704.

## Author contributions

X.W.F., Z.P.H. and Z.F.L. conceived the research project. Z.F.L., H.Z. and J.T.G. synthesized the crystals and characterized the structure. Z.F.L. and H.Z. performed the Lorentz phase microscopy, magnetic force microscopy, and magneto-transport measurements. G.Q.L. performed the first-principles calculations. Z.F.L. performed the in-situ

optical Lorentz phase microscopy. Z.F.L. and Q.P.W. did the micromagnetic simulation. The manuscript was drafted by Z.F.L., Z.P.H., Y.M.Z. and X.W.F. with contributions from H.Z., G.Q.L., J.T.G., Q.P.W., Y.D., Y.H., X.G.H., C.L., M.H.Q., X. S., R.C.Y., X.S.G., Z.M.L. and J.M.L. All the authors contributed to the discussion and revision of the manuscript.

## Competing interests

The authors declare no competing interests.
