## [Peer Review File · Nature Communications]

Room-temperature sub-100 nm Néel-type skyrmions in non-stoichiometric van der Waals ferromagnet Fe_{3-x}GaTe₂ with ultrafast laser writabilityREVIEWER COMMENTS

Reviewer #1 (Remarks to the Author):

This paper is well written and presented a new phase, Fe-deficiency $\text{Fe}_{3-x}\text{GaTe}_2$, observing Néel-type skyrmions. I recommend for publication with minor revision.

-In terms of Fe composition, EDS is never a good quantitative way to determine the composition. The accuracy will be improved if the composition of perfect Fe_3GaTe_2 crystal flake is used as reference to compare with the Fe-deficient flake using the same experimental conditions.

-The space group changed to non-inversion symmetry based on the XRD and HAADF-STEM image. It will be more convincing if additional data using CBED to confirm the space group.

-The Fe columns in Fig.1f are not really clear. The atomic columns do not look clean. A better clear HAADF-STEM image will be better that shows better Fe Columns. Would the author claim that the Fe deficiency occurs only at those FeII positions?

-Can the author control the Fe content?

Reviewer #2 (Remarks to the Author):

Reviewer comments on “Room-temperature sub-100 nm Néel-type skyrmions in non-stoichiometric van der Waals ferromagnet $\text{Fe}_{3-x}\text{GaTe}_2$ with ultrafast laser writability”

In this work, the authors demonstrate the presence of Néel skyrmions in FGT due to a DMI tied to broken inversion symmetry from Fe vacancies. Additionally, the authors demonstrate the ability to drive the phase change optically, by locally heating the sample in-situ with a 520nm pulsed laser during LTEM measurement.

While the material parameters are marginally higher than previously reported in 2D vdW skyrmion systems, it seems to me that the novelty of this work comes from the mechanism/crystallography, which should be better described. The physics of the material is the new discovery here, as the material properties are almost the same as existing systems. I think this work needs to be more motivated by, and discuss, the physics of the system, rather than just experimental observations.

In, for example, $(\text{Fe}_{0.5}\text{Co}_{0.5})_3\text{GeTe}_2$:

- The presence of Neel skyrmions and THE of approximately the same value has previously been reported (Zhang et al. Room-temperature skyrmion lattice in a layered magnet $(\text{Fe}_{0.5}\text{Co}_{0.5})_5\text{GeTe}_2$. Sci. Adv.8,eabm7103(2022).)

- The ordering of the skyrmion lattice has previously been reported (Meisenheimer et al. Ordering of room-temperature magnetic skyrmions in a polar van der Waals magnet. Nat Commun 14, 3744 (2023).)

- A similar phase diagram and thickness dependence has been reported (Zhang et al. Room-temperature skyrmion lattice in a layered magnet $(\text{Fe}_{0.5}\text{Co}_{0.5})_5\text{GeTe}_2$. Sci. Adv.8,eabm7103(2022).)

- Approximately the same Curie temperature and similar mechanism (ordering of empty Fe sites) has been reported (Zhang et al. A room temperature polar magnetic metal. Phys. Rev. Materials 6, 044403 (2022).)

And while the in-situ measurement is new, its value would come from actual dynamical measurements- by doing just quasistatic quenching, it doesn't seem like there is functionally any difference between just T,B cycling without the optical pump (Meisenheimer et al. Ordering of room-temperature magnetic skyrmions in a polar van der Waals magnet. Nat Commun 14, 3744 (2023), Zhang et al. Room-temperature skyrmion lattice in a layered magnet (Fe_{0.5}Co_{0.5})₅GeTe₂. Sci. Adv. 8, eabm7103(2022)., Zhang et al. Room-temperature skyrmion lattice in a layered magnet (Fe_{0.5}Co_{0.5})₅GeTe₂. Sci. Adv. 8, eabm7103(2022).) Especially so because you motivate the experiment from the perspective of "ultrafast writing of skyrmions" (L 104, L 313).

More specifically, there needs to be more discussion on the mechanism of the DMI. A global parameter implies that the empty Fe sites are ordering? DFT is used to simulate the value of DMI, but is the relaxed structure comparable? Does this value match with what is measured (does it give skyrmions/domains of similar size)? What does the anisotropy of the parent compound look like?

Having an in-plane P, to my knowledge, separates this from the existing work, but the parameters are ultimately largely the same? How does the directionality of B change the shape of the skyrmions? It seems like the interaction with D should be unique.

Why is there such a large variance in the sizes of the skyrmions? this is also different to what is generally reported. In fact, it almost looks bimodal in many images.

There needs to be more interpretation of the results and tying back to a structure-property relation for me to be comfortable recommending this paper.

Smaller notes:

You mix cubic and hexagonal coordinates- since the system is hexagonal, you should make this consistent (e.g. [0001] instead of [001], L 138)

You need to soften some statements in the introduction- e.g. rapid thermal annealing is not going to "revolutionize skyrmion logic" (L103), these processes have been around for a while and, additionally, are not particularly chip compatible.

Reviewer #3 (Remarks to the Author):

General Comment: The authors investigate a non-stoichiometric room temperature magnet Fe_{2.86}GaTe₂ crystal where the Fe vacancies induce the formation of DMI by spatial inversion symmetry breaking. Such an in-plane isotropic DMI brings about RT Néel-type skyrmions, and the size of the skyrmions can be regulated by the sample thickness and the external magnetic field. The dynamic writing process of RT skyrmions in Fe_{2.86}GaTe₂ flakes enhances the potential application of spintronic devices.

The paper is timely and of interest.

Comment 1: The Methods section describes the process for obtaining a Fe_{2.86}GaTe₂ single crystal, which was achieved directly by precisely controlling the initial molar ratio of the

powder mixtures and growing conditions. I am inquiring about the method of determining the optimum molar ratio in this work? by experiment or theoretical calculation. And what is the advantage of using iron deficiency as a means of breaking the centrosymmetric structure compared to other methods, such as elemental doping (Ref. 31)?

Comment 2: Please check Supplementary Note 1 and Table S1 to verify the value of non-stoichiometric $\text{Fe}_{3-x}\text{GaTe}_2$.

Comment 3: “Moreover, the size of the skyrmions decreases as the sample thickness becomes thinner, and field-free sub-100 nm skyrmions can be obtained at RT when the thickness falls below a threshold ranging from 40 to 60 nm.” This value is significantly lower compared to other vdW magnets based on iron, as summarised in Fig. 4c. This could be attributed to the competition among DMI, magnetic dipolar interaction, and magnetic anisotropy. Could you please provide further explanation for the presence of sub-100 nm RT skyrmions in $\text{Fe}_{2.86}\text{GaTe}_2$ nanoflakes?

Comment 4: The authors should stress what is the difference between the Fe_i and Fe_{ii} , as this will contribute to understanding why the deviation happened on the Fe_{ii} atoms.

Comment 5: Could the authors present the X-ray diffraction pattern of the $\text{Fe}_{2.86}\text{GaTe}_2$ single crystal?

Comment 6: For most of iron-based ternary tellurides, nanoflakes tend to degrade easily in air. How did the authors avoid degradation during the whole measurements?

Comment 7: The authors should strive for consistency across various measurements. A sample with a thickness of 250 nm was selected for studying magneto-transport signatures, while sub-100 nm RT skyrmions were observed in a thickness range of 40 to 60 nm.

Comment 8: The manuscript should be improved by a careful reading. There are several corrections to be made. For example:

- Just below Fig. 2: 'from +6T to 6T'
- Below Fig. S8: ' Sample Morphology'

Response to reviewers' comments

Dear Referees:

We would like to thank you for your careful review and valuable comments about this paper, which have helped us tremendously to improve this revised manuscript. We have carefully addressed all the concerns in this revised manuscript. Moreover, the point-by-point list of responses to your concerns and comments is clarified as follows.

Response to the Report of Referee A

Referee A's General Comment: This paper is well written and presented a new phase, Fe-deficiency $\text{Fe}_{3-x}\text{GaTe}_2$, observing Néel-type skyrmions. I recommend for publication with minor revision.

Author's reply: We sincerely thank the referee for careful reading of our manuscript and for recommending our manuscript to be published in Nature Communications after minor revision. The valuable suggestions and comments provided by the referee are greatly helpful to improve our manuscript. Below we answer the referee's questions and comments in a point-by-point basis. We hope the referee will be satisfied with the revised manuscript as well as our responses.

Referee A's Comment 1: In terms of Fe composition, EDS is never a good quantitative way to determine the composition. The accuracy will be improved if the composition of perfect Fe_3GaTe_2 crystal flake is used as reference to compare with the Fe-deficient flake using the same experimental conditions.

Author's reply: We highly appreciate the referee's suggestions to enhance the accuracy of composition determination. In order to obtain fully stoichiometric Fe_3GaTe_2 , we have systematically grown a series of $\text{Fe}_{3-x}\text{GaTe}_2$ single crystals by varying the Fe content in the raw material composition, utilizing a Te-flux method. Subsequently, comprehensive energy-dispersive X-ray spectroscopy (EDS) mapping was conducted on the cleaved surfaces of these crystals to determine their chemical composition. To ensure the reliability of the EDS results, mapping was carried out at four distinct areas for each sample. Utilizing the Ga ratio as the normalization factor and ignoring the variation of

Te content, their corresponding chemical composition were established. Table R1 provides a comprehensive overview of the raw material composition and the final crystal composition. It is clearly observed that when the raw Fe ratios fall below 0.8, the $\text{Fe}_{3-x}\text{GaTe}_2$ phase cannot be formed. Instead, a mixture of phases, including GaTe and Ga_2Te_3 , is produced. When the raw Fe ratios are equal to or greater than 0.9, $\text{Fe}_{3-x}\text{GaTe}_2$ single crystals are crystallized, with the Fe content in these single crystals increasing proportionally with the raw Fe ratios. However, an increase in the raw Fe ratios to 1.3 results in the formation of FeTe phases. As surmised in Table R1, we find that the chemical formulas for the crystals with the minimum and maximum Fe content correspond to $\text{Fe}_{2.84\pm 0.05}\text{GaTe}_2$ and $\text{Fe}_{2.96\pm 0.02}\text{GaTe}_2$, respectively. This result indicates that Fe vacancies always exist in the single crystals synthesized using a Te-flux method. The formula of the later one, i.e. $\text{Fe}_{2.96\pm 0.02}\text{GaTe}_2$, is quite close to that of the perfect stoichiometric Fe_3GaTe_2 crystal, and can be regarded as the reference to improve the accuracy suggested by the referee. In the previous version of our manuscript, we reported the observation of Néel-type skyrmions in the $\text{Fe}_{3-x}\text{GaTe}_2$ single crystals synthesized with a raw Fe ratio of 0.9. To highlight the existence of Fe vacancies, the chemical formula was denoted as $\text{Fe}_{2.86}\text{GaTe}_2$, which corresponds to the minimum Fe content determined by the EDS mapping. In the revised manuscript, to enhance the accuracy of the chemical formula, an error bar has been added by summarizing the EDX results obtained at different areas, and the chemical formula is denoted as $\text{Fe}_{2.84\pm 0.05}\text{GaTe}_2$ (see Page 3, Lines 118-121 in the main text). Simultaneously, in accordance with the referee's suggestion, the EDX results of the crystals with other Fe content and associated analysis are also presented in Supplementary Note 1 and Table S1 for reference.

Table R1. Summary of the raw material composition and the final product for the growth of $\text{Fe}_{3-x}\text{GaTe}_2$ samples using the self-flux method.

Molar ratio of Fe: Ga: Te	Mass of Fe (g)	Mass of Ga (g)	Mass of Te (g)	Product
0.6:1:2	0.9349	1.9452	7.1200	GaTe, Ga_2Te_3
0.7:1:2	1.0739	1.9153	7.0107	
0.8:1:2	1.2088	1.8864	6.9048	
0.9:1:2	1.3397	1.8583	6.8020	$\text{Fe}_{2.84\pm 0.05}\text{GaTe}_2$
1.0:1:2	1.4667	1.8310	6.7023	$\text{Fe}_{2.91\pm 0.04}\text{GaTe}_2$
1.1:1:2	1.5900	1.8046	6.6054	$\text{Fe}_{2.95\pm 0.03}\text{GaTe}_2$
1.2:1:2	1.7099	1.7789	6.5113	$\text{Fe}_{2.96\pm 0.02}\text{GaTe}_2$
1.3:1:2	1.8263	1.7539	6.4198	FeTe

Referee A's Comment 2: The space group changed to non-inversion symmetry based on the XRD and HAADF-STEM image. It will be more convincing if additional data using CBED to confirm the space group.

Author's reply: We agree with the referee that Convergent Beam Electron Diffraction (CBED) is an efficient approach to characterize the non-inversion symmetry of the crystal structure. Following the referee's suggestion, we have performed CBED measurements with the electron beam injected along the $[11\bar{2}0]$ zone axis, which is the only zone axis that allows for directly observing significant displacement of Fe_{ii} columns. However, the obtained diffraction disks are seriously overlapped (as shown in Fig. R1a), despite various optimizations, including tuning the camera length, electron beam size, and beam-convergence angle α . Therefore, instead of utilizing CBED, we have performed selected area electron diffraction (SAED) measurements along both the $[11\bar{2}0]$ and $[10\bar{1}0]$ zone axes to confirm the non-inversion structural symmetry, as shown in Figs. R2a and R2b. It is clearly observed that the SAED patterns exhibit a series of $(000l)$ diffraction patterns, such as $(000\bar{5})$, $(000\bar{1})$, (0003) and (0007) . However, the simulations demonstrate that the odd l ($l = 2n + 1$) values of $(000l)$ diffractions are allowed for the non-centrosymmetric space group $P3m1$, but forbidden for the centrosymmetric space group $P6_3/mmc$ (refer to Figs. R2c-f and Fig. R2 note). Thus, the presence of odd l values in the $(000l)$ diffractions confirms that the crystal structure of the $\text{Fe}_{3-x}\text{GaTe}_2$ belong to the non-centrosymmetric $P3m1$ space group, rather than the centrosymmetric $P6_3/mmc$. In the revised manuscript, the SAED patterns and associated discussions are presented in the Supplementary Fig. S9 and the main text (see Page 7, Lines 179-184 in the main text), as also shown below.

“Moreover, we also simulated the selected area electron diffraction (SAED) along the $[11\bar{2}0]$ and $[10\bar{1}0]$ zone axes. The presence of odd l values in the $(000l)$ diffractions, evident in both simulated and experimental SAED results (Fig. S9), further confirms that the crystal structure of the $\text{Fe}_{2.84\pm 0.05}\text{GaTe}_2$ belong to the non-centrosymmetric $P3m1$ space group, rather than the centrosymmetric $P6_3/mmc$ structure of Fe_3GaTe_2 .”

Fig. R1. **a** Experimental CBED patterns of $\text{Fe}_{2.84\pm 0.05}\text{GaTe}_2$ from $[11\bar{2}0]$ direction. **b** Ray diagrams showing how increasing the C2 aperture size causes the CBED pattern to change from one in which individual disks are resolved to one in which all the disks overlap.

Fig. R2. **a, b** SAED (Selected Area Electron Diffraction) along the $[11\bar{2}0]$ and $[10\bar{1}0]$ zone axes. **c, d** Simulated SAED patterns based on the XRD refined non-centrosymmetric structure of $\text{Fe}_{3-x}\text{GaTe}_2$. **e, f** Simulated SAED patterns based on the perfect centrosymmetric structure of Fe_3GaTe_2 .

Fig. R2 note: In Figs. R2a and R2b, the experimental SAED were obtained from FIB-prepared $\text{Fe}_{2.84\pm 0.05}\text{GaTe}_2$ lamella along the $[11\bar{2}0]$ and $[10\bar{1}0]$ zone axes, respectively. Generally, the odd l ($l = 2n + 1$) values of $(000l)$ and $(h\bar{h}2\bar{h}l)$ diffractions are allowed for the non-centrosymmetric space group $P3m1$, but forbidden in a centrosymmetric space group $P6_3/mmc$. To further confirm the differences, we

simulated SAED patterns based on the XRD refined non-centrosymmetric structure of $\text{Fe}_{3-x}\text{GaTe}_2$ (space group $P3m1$, with Fe_{ii} deviation), which align exceptionally well with the experimental results (Fig. R2c and R2d). Instead, a perfect centrosymmetric structure of Fe_3GaTe_2 (space group $P6_3/mmc$, without Fe_{ii} deviation), lacks $(000l)$ and $(h\bar{h}2\bar{h}l)$ diffraction patterns for the odd values of l (Fig. R2c and R2d).

Referee A's Comment 3: The Fe columns in Fig.1f are not really clear. The atomic columns do not look clean. A better clear HAADF-STEM image will be better that shows better Fe Columns. Would the author claim that the Fe deficiency occurs only at those Fe_{II} positions?

Author's reply: We highly appreciate the referee's comments. To provide a clearer view of the Fe_{i} and Fe_{ii} columns, we have further acquired improved HAADF- and ABF-STEM images of the $\text{Fe}_{2.84\pm 0.05}\text{GaTe}_2$ sample along the $[11\bar{2}0]$ zone axis, as shown in Fig. R3. To highlight the detailed information about the Fe_{ii} columns, a magnified image was derived from the enclosed region of the ABF-STEM image, as shown in Fig. R4a. For a quantitative determination of the deviation of the Fe_{ii} atoms, we focused on the region marked by the blue rectangles (left panel of Fig. R4a) comprising Te- Fe_{ii} -Te atoms. We then vertically integrated the corresponding imaging intensity line profile (Fig. R4b). By referencing the center of the two Te atoms, the deviation of the Fe_{ii} atom towards the $-c$ direction was determined to be -0.20 \AA . Utilizing the same procedure, we surveyed an area of 2×17 unit cells, yielding an average Fe_{ii} deviation of $-0.16 \pm 0.06 \text{ \AA}$.

Additionally, we observed that the image intensity of $\text{Fe}_{\text{i-a}}$ above Fe_{ii} is weaker than that of $\text{Fe}_{\text{i-b}}$ below Fe_{ii} , as evident in the imaging intensity line profile of $\text{Fe}_{\text{i-a}}$ - Fe_{ii} - $\text{Fe}_{\text{i-b}}$ atoms in Fig. R4c. Since imaging intensity is generally proportional to the number of projected atoms [*PNAS* **107.26 (2010): 11682-11685**], the contrast difference between $\text{Fe}_{\text{i-a}}$ and $\text{Fe}_{\text{i-b}}$ indicates asymmetric site occupations, suggesting a small quantity of Fe vacancies in the $\text{Fe}_{\text{i-a}}$ site. In the previous version of our manuscript, we emphasized the Fe deficiency at Fe_{ii} sites with an occupancy ratio of 0.8467. Upon re-evaluating the results of the refined single-crystal XRD, we discovered that the lower $\text{Fe}_{\text{i-b}}$ sites are nearly fully occupied, while the higher $\text{Fe}_{\text{i-a}}$ sites have an occupancy ratio of 0.9688. Thus, the Fe deficiency occurs not only at the Fe_{ii} positions, but also at some $\text{Fe}_{\text{i-a}}$ positions.

In the revised manuscript, we have added the associated discussions on Fe_{ii} deviations, Fe_{ii} vacancies and the asymmetric Fe_{i} vacancies (refer to Page 7, Lines 187-

198 and Page 6, Lines 154-159 in the main text) into the main text and Supplementary Figs. S4 and S5, as also shown below.

“However, the HAADF image along the $[11\bar{2}0]$ zone axis (**Fig. 1f**) reveals that Fe_{ii} atoms deviate clearly from the center position of the Te slices along the c -axis, which is also supported by the annular bright-field (ABF-STEM) image in **Fig. S4**. By referencing the center of the two Te atoms in a magnified ABF-STEM image (**Fig. S5**), an averaged Fe_{ii} deviation is calculated as $-0.16 \pm 0.06 \text{ \AA}$ over an area of 2×17 unit cells (see Supplementary **Note 2**).”

“In comparison to the stoichiometric Fe_3GaTe_2 with a centrosymmetric structure, the presence of Fe deficiency in $\text{Fe}_{2.84 \pm 0.05}\text{GaTe}_2$ should exert a pivotal influence on the Fe_{ii} deviation for the asymmetric structure. Our refined single-crystal XRD indicates that Fe deficiency is predominantly concentrated at the Fe_{ii} positions with an occupancy ratio of 0.8467. Additionally, the upper-layer $\text{Fe}_{\text{i-a}}$ sites have an occupancy ratio of 0.9688, while the under-layer $\text{Fe}_{\text{i-b}}$ sites are nearly fully occupied (**Fig. 1h**). As observed in the line profile of $\text{Fe}_{\text{i-a}}$ and $\text{Fe}_{\text{i-b}}$ atoms in the ABF-STEM image (**Fig. S5c**), it is apparent that the image intensity of $\text{Fe}_{\text{i-a}}$ above Fe_{ii} is weaker than that of $\text{Fe}_{\text{i-b}}$ below Fe_{ii} . Since the ABF imaging intensity is generally proportional to the number of projected atoms⁴¹, the contrast difference between $\text{Fe}_{\text{i-a}}$ and $\text{Fe}_{\text{i-b}}$ indicates asymmetric site occupations, suggesting a small quantity of Fe vacancies in the $\text{Fe}_{\text{i-a}}$ site, which is consistent with the results of single-crystal XRD.”

Fig. R3. **a** High-angle annular dark-field scanning transmission electron microscopy (HAADF-STEM) image and **b** annular bright-field scanning transmission electron microscopy (ABF-STEM) image of the $\text{Fe}_{2.84 \pm 0.05}\text{GaTe}_2$ sample along the $[11\bar{2}0]$ zone axis.

Fig. R4. **a** Magnified ABF-STEM image of the single $\text{Fe}_{2.84\pm 0.05}\text{GaTe}_2$ layer. **b** Integrated imaging intensity line profile along the c -axis within the area marked by the Te-Fe_{ii}-Te atoms in the blue rectangles. The red region indicates the Fe_{ii} deviation from the centers of Te-Te atoms. **c** Integrated imaging intensity line profile of Fe_{i-a}-Fe_{i-b} atoms.

Referee A's Comment 4: Can the author control the Fe content?

Author's reply: We sincerely appreciate the insightful comments provided by the referee. Following the comments, we have further systematically grown a series of $\text{Fe}_{3-x}\text{GaTe}_2$ single crystals by varying the Fe content in the raw material composition, utilizing a Te-flux method. A comprehensive overview of the raw material composition and the final product is outlined in Table R1 (see Author's reply to Referee A' Comment 1). It is clearly demonstrated that controlling the Fe content in the final crystals is achievable by varying the raw Fe ratio. Specifically, when the raw Fe ratio falls below 0.8, the $\text{Fe}_{3-x}\text{GaTe}_2$ phase cannot be formed. Instead, a mixture of phases, including GaTe and Ga_2Te_3 phases, is produced. In contrast, when the raw Fe ratio is equal to or greater than 0.9, $\text{Fe}_{3-x}\text{GaTe}_2$ single crystals can be crystallized, with the Fe content in these single crystals increasing proportionally with the raw Fe ratio. However, an increase in the raw Fe ratio to 1.3 results in the formation of FeTe phase. In the revised manuscript, we have incorporated discussions on controlling the Fe content of the crystals into both the main text (refer to Page 5, Lines 110-120) and Supplementary information (refer to Supplementary Note 1 and Table S1), providing a more comprehensive understanding of the growth conditions and their impact on the final composition of the single crystals.

“In order to control the Fe content, we systematically grew a series of $\text{Fe}_{3-x}\text{GaTe}_2$ single crystals by varying the Fe content in the raw material composition, utilizing a Te-flux method (see **Methods** section and Supplementary **Note 1**). To determine the chemical composition of the as-grown crystals, energy dispersive X-ray spectroscopy (EDX) analyses were conducted on the surfaces of $\text{Fe}_{3-x}\text{GaTe}_2$ nanoflakes (**Fig. 1a**) that were exfoliated and placed onto the Si_3N_4 membrane (see **Methods**). The ratio of raw materials and the corresponding final crystal composition are listed in **Table S1**, Supplementary **Fig. S1** and **Fig. 1b**. We found that the Fe deficiencies always exist in these crystals, while the minimum and maximum Fe contents correspond to $\text{Fe}_{2.84\pm 0.05}\text{GaTe}_2$ and $\text{Fe}_{2.96\pm 0.02}\text{GaTe}_2$, respectively. This result implies the feasibility of inducing Fe deficiency in the samples. To highlight the existence of Fe vacancies, the subsequent studies were focused on the minimum Fe content sample $\text{Fe}_{2.84\pm 0.05}\text{GaTe}_2$.”

Response to the Report of Referee B

Referee B's General Comment: In this work, the authors demonstrate the presence of Néel skyrmions in FGT due to a DMI tied to broken inversion symmetry from Fe vacancies. Additionally, the authors demonstrate the ability to drive the phase change optically, by locally heating the sample in-situ with a 520 nm pulsed laser during LTEM measurement. While the material parameters are marginally higher than previously reported in 2D vdW skyrmion systems, it seems to me that the novelty of this work comes from the mechanism/crystallography, which should be better described. The physics of the material is the new discovery here, as the material properties are almost the same as existing systems. I think this work needs to be more motivated by, and discuss, the physics of the system, rather than just experimental observations.

In, for example, $(\text{Fe}_{0.5}\text{Co}_{0.5})_3\text{GeTe}_2$:

- The presence of Neel skyrmions and THE of approximately the same value has previously been reported (Zhang et al. Room-temperature skyrmion lattice in a layered magnet $(\text{Fe}_{0.5}\text{Co}_{0.5})_5\text{GeTe}_2$. *Sci. Adv.* 8, eabm7103(2022).)
- The ordering of the skyrmion lattice has previously been reported (Meisenheimer et al. Ordering of room-temperature magnetic skyrmions in a polar van der Waals magnet. *Nat Commun* 14, 3744 (2023).)
- A similar phase diagram and thickness dependence has been reported (Zhang et al. Room-temperature skyrmion lattice in a layered magnet $(\text{Fe}_{0.5}\text{Co}_{0.5})_5\text{GeTe}_2$. *Sci. Adv.* 8, eabm7103(2022).)
- Approximately the same Curie temperature and similar mechanism (ordering of empty Fe sites) has been reported (Zhang et al. A room temperature polar magnetic metal. *Phys. Rev. Materials* 6, 044403 (2022).)

Author's reply: We sincerely appreciate the referee's comments regarding more in-depth discussion on the mechanism/crystallography. These valuable comments are of great significance to improve the quality of our manuscript. In the last version of our manuscript, our primary focus is on reporting that the Fe_{ii} vacancies in $\text{Fe}_{3-x}\text{GaTe}_2$ induces the displacement of Fe_{ii} atoms, leading to the breaking of crystal inversion symmetry. These vacancies serve as the source of the Dzyaloshinskii–Moriya interaction, which not only contributes significantly to a room-temperature topological Hall effect but also facilitates the formation of small-sized Néel-type skyrmions with fs laser writability. Following the referee's comments, we have made a major revision to

reinforce the discussion on the mechanism and crystallography. The key points of our accomplishments are outlined below, with more comprehensive discussions provided in subsequent responses to Referee B's Comments 2. We hope the referee will be satisfied with the revised manuscript as well as our responses.

(i) To understand the underlying physics for symmetry breaking on Fe_{ii} sites, we have first conducted a thorough examination, employing improved ABF-STEM images and re-evaluating the single crystal XRD data. Our investigation has revealed that Fe deficiency is predominantly concentrated at the Fe_{ii} positions. Additionally, a minor presence of asymmetric Fe deficiency at the Fe_{i} positions was observed, wherein the upper-layer $\text{Fe}_{\text{i-a}}$ sites exhibit a higher degree of deficiency compared to the lower-layer $\text{Fe}_{\text{i-b}}$ sites.

(ii) Upon establishing the crystal structure, we have conducted structure relaxation based on DFT calculations, considering $\text{Fe}_{\text{i-a}}$ and Fe_{ii} vacancies separately. Our computational results confirm that the asymmetric $\text{Fe}_{\text{i-a}}$ vacancy primarily induces Fe_{ii} deviation towards the $-c$ direction, which results in the symmetry breaking of the crystal, whereas the Fe_{ii} vacancy exerts no influence on Fe_{ii} deviation.

(iii) Furthermore, we explored the correlation between Fe_{ii} deviation values δc ($\text{Fe}_{\text{ii}} - \text{Ga}$) and the DMI constant D based on the DFT calculations. It is found that no DMI is observed (the value of D is equal to zero) in a centrosymmetric structure without Fe_{ii} deviation ($\delta c = 0$). However, once the Fe_{ii} atom deviate from Ga plane ($\delta c < 0$) due to the asymmetric $\text{Fe}_{\text{i-a}}$ vacancy, the inversion symmetry is broken and the value of D increases accordingly as the increase of the Fe_{ii} deviation. It should be noted that the calculated value of D is comparable to that established in experiments, confirming the reliability of our structural model.

We appreciate the referee's recommendation of the excellent work on $(\text{Fe}_{0.5}\text{Co}_{0.5})_5\text{GeTe}_2$ and polar van der Waals magnets. All relevant references have been appropriately cited in the main text to acknowledge the valuable contributions of the prior work in the field.

Referee B's Comment 1: And while the in-situ measurement is new, its value would come from actual dynamical measurements- by doing just quasistatic quenching, it doesn't seem like there is functionally any difference between just T, B cycling without the optical pump (Meisenheimer et al. Ordering of room-temperature magnetic skyrmions in a polar van der Waals magnet. Nat Commun 14, 3744 (2023); Zhang et al. Room-temperature skyrmion lattice in a layered magnet $(\text{Fe}_{0.5}\text{Co}_{0.5})_5\text{GeTe}_2$. Sci. Adv.8,eabm7103(2022)). Especially so because you motivate the experiment from the

perspective of “ultrafast writing of skyrmions” (L 104, L 313).

Author’s reply: We are grateful for the referee's positive feedback and suggestions regarding the in-situ measurements. Femtosecond laser (fs) control of topological magnetic structures is a promising and still relatively unexplored field, involving complex physical processes such as ultrafast demagnetization [**Nature Communications** **14.1 (2023): 1378**], optically induced magnetism [**Physical Review Letters** **125.26 (2020): 267205**], optical-pumped spin dynamics, all-optical magnetization reversal, and more. It is widely known that, magnetic skyrmions in two-dimensional materials are often metastable and typically require temperature-magnetic field (*T-B*) cycling [**Nature Communications** **13.1 (2022): 3035**; **Nano letters** **22.19 (2022): 7804-7810**], which is time-consuming and energy-intensive. In contrast, fs laser pulse-induced skyrmion writing, based on its unique quenching effect, offers the advantages of fast speed and low energy consumption [**Nature Materials** **20.1 (2021): 30-37**]. Moreover, it also allows for the adjustment of laser spot size and location, enabling selective writing in specific regions [**Nano Letters** **18.11 (2018): 7362-7371**]. Consequently, there has been significant interest in laser-induced change and switching of topological spin textures in recent years, which allows the exploration of metastability and hidden phases of topological spin textures [**Science Advances** **4.7 (2018): eaat3077**].

In further response to the referee's comments, we'd like to show that there is indeed a functional difference between the *T-B* cycling and our in-site fs laser quenching approach. Typically, without a magnetic field, zero-field cooling can only result in the formation of interconnected, relatively long stripe domains, but does not spontaneously lead to the creation of skyrmions (as show in Fig. R5a-c) [**Nature Communications** **13.1 (2022): 3035**]. However, through our measurements with varying laser pulse fluences, we have identified the possibility of achieving skyrmion writing under zero magnetic field conditions. As depicted in Fig. R5a and R5d, under the condition of a single laser pulse fluence of 1.3 mJ/cm², the stripe domains within the yellow box merely exhibit domain wall movement after the laser pulse. In Fig. R5b and R5e, when we increase the single laser pulse fluence to 9.4 mJ/cm², the stripe domains become narrower and shorter, with some regions breaking to form skyrmions (highlighted in the red box). However, as we increase the single laser pulse fluence to 11 mJ/cm², stripe domains are formed without skyrmions (Fig. R5c and R5f). These in-situ laser fluence-dependent experiments indicate that a hybrid state with coexisting stripes and skyrmions is achievable without magnetic field. Currently, all the reported articles on fs laser-induced skyrmion writing have required external magnetic field assistance

[Nature Communications 14.1 (2023): 1378, Nature Materials 20.1 (2021): 30-37, Nano Letters 18.11 (2018): 7362-7371]. Exploring the possibility of all-optical writing of skyrmions without magnetic field is an exciting prospect. We anticipate that future work would allow for a more in-depth exploration of this intriguing possibility. In response, we have included these laser fluence-dependent experiments in the supplementary materials (Fig. S25), and added a simplified description based on the experimental observation in the main text (refer to Page 15, Lines 437-443), as also shown below.

“Furthermore, we conducted fluence-dependent laser pulse excitation without magnetic field (see Fig. S25 and Supplementary Note 9). After a single laser pulse with fluence of 1.3 mJ/cm^2 , stripe domains show slightly domain wall movement. Upon increasing to 9.4 mJ/cm^2 , a hybrid state with both stripes and skyrmions are formed, while at 11 mJ/cm^2 only stripe domains observed. This indicates that a higher fluence of fs laser can completely demagnetize the sample during the laser-writing process, regardless of the initial magnetic state.”

Fig. R5. a, b, c. Ground states of the stripe domains obtained through zero-field cooling. d, e, f. Magnetic domain states after a single fs laser pulse with the fluence of 1.3 mJ/cm^2 , 9.4 mJ/cm^2 , and 11 mJ/cm^2 , respectively. The red boxes indicated isolated skyrmions after a single fs laser pulse.

Referee B’s Comment 2: More specifically, there needs to be more discussion on the mechanism of the DMI. A global parameter implies that the empty Fe sites are ordering? DFT is used to simulate the value of DMI, but is the relaxed structure comparable? Does this value match with what is measured (does it give skyrmions/domains of similar size)? What does the anisotropy of the parent compound look like?

Author's reply: We sincerely thank the referee for careful reading of our manuscript. These valuable suggestions and comments are greatly helpful for us to explore a more in-depth physical picture on the mechanism of the DMI. In response to these comments, we have supplemented additional structural characterizations and first-principles calculations, revealing that the asymmetric Fe_i vacancies are the primary cause of the Fe_{ii} deviations. Furthermore, the computed DMI value increases with the larger Fe_{ii} deviations, showing comparable values with the experiment results. Additionally, we have compared the magnetic anisotropy of samples with different compositions. More detailed discussions are listed below:

1. Responses to "A global parameter implies that the empty Fe sites are ordering?":

Our earlier version of manuscript has reported Fe_{ii} vacancies and Fe_{ii} deviations in $\text{Fe}_{3-x}\text{GaTe}_2$. As suggested by the referee, it's crucial to understand the order of empty Fe sites (Fe_i and Fe_{ii}), which will facilitate us to explain why deviations happened on Fe_{ii} sites. In order to provide a better clear view of the Fe_i and Fe_{ii} columns, we have acquired improved ABF-STEM images of the $\text{Fe}_{2.84\pm 0.05}\text{GaTe}_2$ sample along the $[11\bar{2}0]$ zone axis. As depicted in Fig. R6a, the magnified ABF-STEM image of the single layer distinctly shows the Fe_i and Fe_{ii} atoms adjacent to the Ga atoms. For a quantitative determination of the deviation of the Fe_{ii} atoms, we focused on the region marked by the left blue rectangle comprising the Te- Fe_{ii} -Te atoms in Fig. R6a. We then vertically integrated the corresponding imaging intensity line profile (Fig. R6b). By referencing the center of the two Te atoms, the deviation of the Fe_{ii} atom towards the $-c$ direction was determined to be -0.20 \AA . Utilizing the same procedure, we surveyed an area of 2×17 unit cells, yielding an average Fe_{ii} deviation of $-0.16 \pm 0.06 \text{ \AA}$.

Additionally, we observed that the image intensity of Fe_{i-a} above Fe_{ii} is weaker than that of Fe_{i-b} below Fe_{ii} , as evident in the imaging intensity line profile of Fe_{i-a} - Fe_{i-b} atoms indicated by the right blue rectangle in Fig. R6c. Since imaging intensity is generally proportional to the number of projected atoms [PNAS 107.26 (2010): 11682-11685], the contrast difference between Fe_{i-a} and Fe_{i-b} indicates asymmetric site occupations, suggesting a small quantity of Fe vacancies in the Fe_{i-a} site. In the previous version of our manuscript, we emphasized the Fe deficiency at Fe_{ii} sites with an occupancy ratio of 0.8467. Upon re-evaluating the results of the refined single-crystal XRD, we discovered that the lower Fe_{i-b} sites are nearly fully occupied, while the higher Fe_{i-a} sites have an occupancy ratio of 0.9688. Therefore, we claim that the Fe deficiency occurs not only at the Fe_{ii} positions, but also at some Fe_{i-a} positions.

Having established the structure model of the Fe vacancies, we conducted structure

relaxation in DFT calculations to analyze the impact of Fe_{i-a} and Fe_{ii} vacancies on Fe_{ii} deviation. Starting with a perfect centrosymmetric Fe_3GaTe_2 lattice, we constructed a 2×2 supercell encompassing a total of four molecular layers. Within the bottommost layer, we systematically introduced three scenarios: no vacancy, Fe_{i-a} vacancy, and Fe_{ii} vacancy. Lattice relaxations were performed independently for the three distinct supercells. In order to illustrate the Fe vacancies and further compare the alterations in Fe_{ii} chemical bonding, we presented the electron density of Fe_{3-x}Ga atoms within the bottommost layer (Fig. R7). Specific computational details can be found in Fig. R7 note.

Fig. R7a and R7b illustrate the scenario with no Fe vacancy. The electron density (colored in yellow) strongly overlaps between Fe_{ii} -Ga atoms, forming the Fe_{ii} -Ga honeycomb lattice plane with robust chemical bonding (highlighted by black dashed lines). Simultaneously, the Fe_{i-a} and Fe_{i-b} dimers are located at the center of the Fe_{ii} -Ga honeycomb lattice but do not bond with Fe_{ii} -Ga atoms. Consequently, the chemical bonding is mirror-symmetric along the Fe_{ii} -Ga plane, leading to the absence of Fe_{ii} displacements. Thus, perfect Fe_3GaTe_2 exhibits a centrosymmetric crystal structure with $c \rightarrow -c$ mirror symmetry.

Fig. R7c and R7d illustrate the scenario with Fe_{ii} vacancy. Although Fe_{ii} vacancies cause deformation of the Ga electron density within the ab plane, there is still no overlapping with of Fe_{i-a} and Fe_{i-b} in the c direction. This observation indicates that the remaining Fe_{ii} primarily forms bonds with Ga in the ab plane, with no displacement observed in the c direction.

Fig. R7e and R7f illustrate the scenario with Fe_{i-a} vacancy. Apart from the strongly bonded Fe_{ii} -Ga honeycomb lattice, there is additional electron-density overlapping among the lower Fe_{i-b} atom and its three nearest Fe_{ii} atoms, while no overlapping between Fe_{i-b} and its three nearest Ga atoms. Consequently, the newly formed chemical bonding (highlighted by black dashed line) will drag the Fe_{ii} atoms displacing downwards to the lower Fe_{i-b} atom. Notably, the determined deviation between nearest Fe_{ii} and Ga atoms in the relaxed structure is -0.0554 \AA . This value aligns well with the analysis from ABF-STEM image, confirming the reliability of our DFT model. Based on the DFT results above (Fig. R7), we can conclude that the asymmetric vacancy of Fe_{i-a} induces a displacement of Fe_{ii} atoms towards the $-c$ direction, resulting in the spatial inversion symmetry breaking.

In the revised version of the manuscript, we have integrated experimental findings and DFT structure relaxations on Fe_{ii} deviations, Fe_{ii} vacancies and the asymmetric Fe_{i-a} vacancies (refer to Page 6, Lines 154-159, Pages 7 and 8, Lines 197-218 in the main text), as also shown below. The corresponding analyses and experimental data are

comprehensively presented in Supplementary Note 3 and Fig. S10.

“However, the HAADF image along the $[11\bar{2}0]$ zone axis (**Fig. 1f**) reveals that Fe_{ii} atoms deviate clearly from the center position of the Te slices along the c -axis, which is also supported by the annular bright-field (ABF-STEM) image in **Fig. S4**. By referencing the center of the two Te atoms in a magnified ABF-STEM image (**Fig. S5**), an averaged Fe_{ii} deviation is calculated as $-0.16 \pm 0.06 \text{ \AA}$ over an area of 2×17 unit cells (see Supplementary **Note 2**).”

“In comparison to the stoichiometric Fe_3GaTe_2 with a centrosymmetric structure, the presence of Fe deficiency in $\text{Fe}_{2.84 \pm 0.05}\text{GaTe}_2$ should exert a pivotal influence on the Fe_{ii} deviation for the asymmetric structure. Our refined single-crystal XRD indicates that Fe deficiency is predominantly concentrated at the Fe_{ii} positions with an occupancy ratio of 0.8467. Additionally, the upper-layer $\text{Fe}_{\text{i-a}}$ sites have an occupancy ratio of 0.9688, while the under-layer $\text{Fe}_{\text{i-b}}$ sites are nearly fully occupied (**Fig. 1h**). As observed in the line profile of $\text{Fe}_{\text{i-a}}$ and $\text{Fe}_{\text{i-b}}$ atoms in the ABF-STEM image (**Fig. S5c**), it is apparent that the image intensity of $\text{Fe}_{\text{i-a}}$ above Fe_{ii} is weaker than that of $\text{Fe}_{\text{i-b}}$ below Fe_{ii} . Since the ABF imaging intensity is generally proportional to the number of projected atoms⁴¹, the contrast difference between $\text{Fe}_{\text{i-a}}$ and $\text{Fe}_{\text{i-b}}$ indicates asymmetric site occupations, suggesting a small quantity of Fe vacancies in the $\text{Fe}_{\text{i-a}}$ site, which is consistent with the results of single-crystal XRD. To assess the influence of $\text{Fe}_{\text{i-a}}$ and Fe_{ii} vacancies on Fe_{ii} deviation, we further conducted first-principles calculations involving structure relaxation under three scenarios: no vacancy, $\text{Fe}_{\text{i-a}}$ vacancy, and Fe_{ii} vacancy (Supplementary **Note 3**). The electron density of Fe_{3-x}Ga atoms is depicted in **Fig. S10** to facilitate a comparison of the alterations in Fe_{ii} chemical bonding: (a) The perfect Fe_3GaTe_2 , with no Fe vacancy, showcases a hexagonally bonded Fe_{ii} -Ga plane. In this arrangement, the centrally positioned $\text{Fe}_{\text{i-a}}$ and $\text{Fe}_{\text{i-b}}$ dimers do not form direct bonds with Fe_{ii} and Ga atoms. Thus, the overall chemical bonding is mirror-symmetric along the Fe_{ii} -Ga plane with no Fe_{ii} deviation. (b) The presence of Fe_{ii} vacancy induces deformation of the Ga electron density within the ab plane. Nevertheless, no bonding is established between the Fe_{ii} -Ga plane and the Fe_{i} atoms, which remain a mirror-symmetric electron density with no Fe_{ii} deviation. (c) In case of $\text{Fe}_{\text{i-a}}$ vacancy, there is additional electron-density overlapping between the lower $\text{Fe}_{\text{i-b}}$ atom and its three nearest Fe_{ii} atoms, while no overlapping between $\text{Fe}_{\text{i-b}}$ and its three nearest Ga atoms. As a result, chemical bonding between the Fe_{ii} and $\text{Fe}_{\text{i-b}}$ atoms induces a substantial Fe_{ii} deviation, with a calculated δc ($\text{Fe}_{\text{ii}} - \text{Ga}$) of about -0.0554 \AA , which compares favorably to the XRD result of -0.0871 \AA . Furthermore, the calculated formation energy value for Fe_{i} vacancy (2.96 eV/Fe) is higher than Fe_{ii} vacancy (2.86 eV/Fe),

indicating that the formation of Fe_{ii} vacancies is more favorable. The above Fe vacancy model and calculated Fe_{ii} deviation align well with the analysis from single-crystal XRD and ABF-STEM image. Therefore, we conclude that the asymmetric vacancy of $\text{Fe}_{\text{i-a}}$ induces a displacement of Fe_{ii} atoms towards the $-c$ direction, which results in the symmetry breaking of the $\text{Fe}_{2.84\pm 0.05}\text{GaTe}_2$ crystal structure.”

Fig. R6. **a** Magnified ABF-STEM image of the single $\text{Fe}_{2.84\pm 0.05}\text{GaTe}_2$ layer. **b** Integrated imaging intensity line profile along the c -axis within the area marked by the Te-Fe_{ii}-Te atoms in the left blue rectangle. The red region indicates the Fe_{ii} deviation from the centers of Te-Te atoms. **c** Integrated imaging intensity line profile of $\text{Fe}_{\text{i-a}}$ and $\text{Fe}_{\text{i-b}}$ atoms in the region indicated by the right blue rectangle.

Fig. R7. The relaxed crystal structure and corresponding electron density of Fe_{3-x}Ga atoms sliced from $\text{Fe}_{3-x}\text{GaTe}_2$ with **a, b** no vacancy, **c, d** Fe_{ii} vacancy and **e, f** $\text{Fe}_{\text{i-a}}$ vacancy. The black dashed line indicates the chemical bonding with electron-density overlapping. The yellow-colored electron densities are shown at the same isosurface value.

Fig. R7 note: In all cases presented, the Vienna ab initio simulation package (VASP) was used with electron-core interactions described by the projector augmented wave method for the pseudopotentials, and the exchange correlation energy calculated with the generalized gradient approximation of the Perdew-Burke-Ernzerhof (PBE) form. The plane wave cutoff energy was 400 eV for all the calculations. In calculating the atomic shifts due to $\text{Fe}_{\text{i-a}}$ and Fe_{ii} vacancies, we used a 2×2 supercell and removed the bottommost $\text{Fe}_{\text{i-a}}$ and Fe_{ii} atoms. The Monkhorst-Pack scheme was used for the Γ -centred $12 \times 12 \times 1$ k-point sampling. All atoms' relaxations were performed until the force become smaller than 0.001 eV/Å for determining the most stable geometries.

2. Response to “DFT is used to simulate the value of DMI, but is the relaxed structure comparable? Does this value match with what is measured (does it give skyrmions/domains of similar size)?”:

The above ABF-STEM analysis has revealed the Fe deficiency is predominantly concentrated at the Fe_{ii} positions, accompanied by a minor deficiency in asymmetric

Fe_{i-a} positions. However, building such a vacancy model with exact occupancy ratio of Fe_{ii} and Fe_{i-a} atoms requires a very large supercell structure, which is beyond the computation capability for DMI calculation. To reduce the computational complexity, the DFT calculation is based on the fixed crystal structure obtained from single-crystal XRD experiments, instead of a fully relaxed crystal structure optimized from DFT calculations. The reasons are described below:

In considering the structural model in Fig. R8, the formation of DMI requires spatial inversion symmetry breaking in upper and lower triangles composed of Fe_i-Fe_{ii}-Te atoms. In quantitative terms, the DMI vector can be expressed as

$$\mathbf{D} = D \cdot (\hat{\mathbf{u}}_{ij} \times \hat{\mathbf{z}}),$$

where D is the DMI constant, \mathbf{u}_{ij} represents the unit vector from Fe_i atom to Fe_{ii} atom, and \mathbf{z} represents the unit vector from magnetic Fe_{ii} atom to heavy Te atom. It can be seen that if Fe_{ii} is located at the center position of Te-Te atoms, where ever Fe_{i-a} or Fe_{i-b} are located, the upper D_1 and lower D_2 vectors would always cancel out with each other. This feature suggests that the spatial inversion symmetry breaking of Fe_{ii} deviations is the primary cause of DMI, while Fe_{i-a} vacancies do not contribute DMI. Thus, we can reasonably assume that atoms are fixed with full occupancy, and ignoring the steps of structure relaxation, which would otherwise necessitate the creation of impractically large supercells in the vacancy model.

In order to compare the DFT calculations with experimental observations, we investigated the relationship between Fe_{ii} deviation value $\delta c(\text{Fe}_{ii} - \text{Ga})$ and DMI constant D . As shown in Fig. R9a and R9b, the crystal structures were built with different Fe_{ii} deviation values $\delta c(\text{Fe}_{ii} - \text{Ga})$, which are fixed during the two-step calculation for DMI values. For more detailed about the DFT calculation, please refer to Fig. R9 note. The calculated D with different $\delta c(\text{Fe}_{ii} - \text{Ga})$ values are shown in Fig. R9c. On the one hand, the minimum $\delta c = 0$ indicate a centrosymmetric structure without Fe_{ii} deviation. And the corresponding DMI value is $D = 0 \text{ mJ/m}^2$, which is consistent with our expectations. On the other hand, the maximum $\delta c = -0.0871 \text{ \AA}$ is equaling to the non-centrosymmetric structure determined by single-crystal XRD. And the corresponding DMI value is $D = 0.91 \text{ mJ/m}^2$. Based on the measured magnetic domain width from LTEM experiments, our previous version of manuscript has extracted the $D = 0.25 \text{ mJ/m}^2$ (see Supplementary Note 5), which falls within the range of DFT calculated D values. In summary, both of the calculated DMI and the Fe_{ii} deviation match well with the experimental value. We have incorporated discussions on DMI calculations in both the main text (refer to Page 9, Lines 246-252, as also shown below),

and Supplementary information (refer to Supplementary Note 4 and Fig. S12).

“Based on the model depicted in the aforementioned illustration, we quantitatively investigated the relationship between the Fe_{ii} deviation value δc ($Fe_{ii} - Ga$) and the DMI constant D (see Supplementary Note 4 and Fig. S12 for details). For a centrosymmetric structure ($\delta c = 0$), the absence of Fe_{ii} deviation yields $D = 0 \text{ mJ/m}^2$. Conversely, a non-centrosymmetric structure ($\delta c = -0.0871 \text{ \AA}$), determined by single-crystal XRD, corresponds to $D = 0.91 \text{ mJ/m}^2$.”

Fig. R8. Schematic illustration of DMI in asymmetric layers viewed from $[11\bar{2}0]$ zone axis. The red arrow D_1 represents the direction of DMI vector in the upper triangle composed of Fe_i - Fe_{ii} -Te, while the blue arrow D_2 represents the lower part in the opposite direction. The black arrow D_{eff} represents the sum of the non-zero DMI vector.

Fig. R9. **a, b** Spin configurations implemented to calculate the DMI for clockwise (CW) and anticlockwise (ACW). **c** The calculated and experimental results for the relationship between the DMI and the Fe_{ii} deviation.

Fig. R9 note: The calculation of the DMI vector involved two steps. First, structural relaxations with fixed $\delta(\text{Fe}_{ii})$ were performed with Gaussian smearing until the forces become smaller than 0.001 eV/\AA . Next, spin-orbit coupling was included in the calculation, and the total energy of the system was determined as a function of the spin configuration as shown in Fig. R9a, and $d_{||}$ equals to $(E_{ACW} - E_{CW})/12$. The DMI constant D was calculated using the equation $D = 3\sqrt{2}d_{||}/(N_F a^2)$, where N_F is the number of atomic layers, a is the lattice constant and $d_{||}$ represents DMI strength. In the second step, the EDIFF is set to 10^{-8} eV and the tetrahedron method with Blöchl corrections was used to get an accurate total-energy. The resulting relationship between D and $\delta c(\text{Fe}_{ii} - \text{Ga})$ is presented in Fig. R9b.

3. Responses to: “What does the anisotropy of the parent compound look like?”

In order to compare the anisotropy of different Fe-content samples, we systematically have grown a series of $\text{Fe}_{3-x}\text{GaTe}_2$ single crystals by varying the Fe content in the raw material composition, utilizing a Te-flux method. To ensure the reliability of the compositions, EDS mapping was carried out at four distinct cleaved surfaces of these crystals. As surmised in Table R2, the chemical formulas for the crystals with the minimum and maximum Fe content correspond to $\text{Fe}_{2.84\pm 0.05}\text{GaTe}_2$ and $\text{Fe}_{2.96\pm 0.02}\text{GaTe}_2$, respectively. This result indicates that Fe vacancies always exist in the single crystals synthesized using a Te-flux method. The chemical formula of

$\text{Fe}_{2.96\pm 0.02}\text{GaTe}_2$ is quite close to that of the parent Fe_3GaTe_2 , which can be regarded at the parent compound for study the anisotropy.

We further measured field dependent magnetization curves for both $\text{Fe}_{2.84\pm 0.05}\text{GaTe}_2$ and $\text{Fe}_{2.96\pm 0.02}\text{GaTe}_2$ samples, as shown in Fig. R10. The easy magnetization direction is along the c axis (out-of-plane), which indicate a typical perpendicular magnetic anisotropy. In comparison with the high Fe content $\text{Fe}_{2.96\pm 0.02}\text{GaTe}_2$ sample, it can be seen that there were no obvious changes on saturation field in easy axis ($B // c$ axis), while slight increment on saturation field ($\leq 2\text{mT}$) in hard axis ($B // ab$ plane). This result indicates that the Fe deficiency has little impact on magnetic anisotropy.

In the revised manuscript, we have incorporated discussions on controlling the Fe content of the crystals, as well as the corresponding field dependent magnetization curves, in both the main text (refer to Page 5, Lines 110-120 and 133-137, as also shown below), Method section (refer to Page 17, Lines 498-500) and Supplementary information (refer to Supplementary Note 1, Table S1 and Fig. S2).

“In order to control the Fe content, we systematically grew a series of $\text{Fe}_{3-x}\text{GaTe}_2$ single crystals by varying the Fe content in the raw material composition, utilizing a Te-flux method (see **Methods** section and **Supplementary Note 1**). To determine the chemical composition of the as-grown crystals, energy dispersive X-ray spectroscopy (EDX) analyses were conducted on the surfaces of $\text{Fe}_{3-x}\text{GaTe}_2$ nanoflakes (**Fig. 1a**) that were exfoliated and placed onto the Si_3N_4 membrane (see **Methods**). The ratio of raw materials and the corresponding final crystal composition are listed in **Table S1**, **Supplementary Fig. S1** and **Fig. 1b**. We found that the Fe deficiencies always exist in these crystals, while the minimum and maximum Fe contents correspond to $\text{Fe}_{2.84\pm 0.05}\text{GaTe}_2$ and $\text{Fe}_{2.96\pm 0.02}\text{GaTe}_2$, respectively. This result implies the feasibility of inducing Fe deficiency in the samples. To highlight the existence of Fe vacancies, the subsequent studies were focused on the minimum Fe content sample $\text{Fe}_{2.84\pm 0.05}\text{GaTe}_2$.”

“Moreover, the field-dependent magnetization curves for the $\text{Fe}_{2.84\pm 0.05}\text{GaTe}_2$ sample with minimum Fe content reveal an out-of-plane easy magnetization direction at room-temperature (**Fig. S2**). These curves exhibit magnetic anisotropy almost identical to that of $\text{Fe}_{2.96\pm 0.02}\text{GaTe}_2$ sample with high Fe content.”

Table R2. Summary of the raw material composition and the final product for the growth of $\text{Fe}_{3-x}\text{GaTe}_2$ samples using the self-flux method.

Molar ratio of Fe: Ga: Te	Mass of Fe (g)	Mass of Ga (g)	Mass of Te (g)	Product
0.6:1:2	0.9349	1.9452	7.1200	GaTe, Ga ₂ Te ₃
0.7:1:2	1.0739	1.9153	7.0107	
0.8:1:2	1.2088	1.8864	6.9048	
0.9:1:2	1.3397	1.8583	6.8020	Fe _{2.84±0.05} GaTe ₂
1.0:1:2	1.4667	1.8310	6.7023	Fe _{2.91±0.04} GaTe ₂
1.1:1:2	1.5900	1.8046	6.6054	Fe _{2.95±0.03} GaTe ₂
1.2:1:2	1.7099	1.7789	6.5113	Fe _{2.96±0.02} GaTe ₂
1.3:1:2	1.8263	1.7539	6.4198	FeTe

Fig. R10. Field dependent magnetization curves for $\text{Fe}_{2.84\pm 0.05}\text{GaTe}_2$ and $\text{Fe}_{2.96\pm 0.02}\text{GaTe}_2$ samples.

Referee B's Comment 3: Having an in-plane P , to my knowledge, separates this from the existing work, but the parameters are ultimately largely the same? How does the directionality of B change the shape of the skyrmions? It seems like the interaction with D should be unique.

Author's reply: We sincerely thank the referee for the valuable comments. Based on the structure model with Fe_{ii} deviation, we conclude the DMI vector D possesses an in-plane isotropic. Fig. R11a schematically illustrates the formation mechanism of DMI viewed from $[11\bar{2}0]$ zone axis, along with the Fe_{ii} deviation induces spatial inversion symmetry breaking in upper and lower triangles composed of $\text{Fe}_i\text{-Fe}_{ii}\text{-Te}$ atoms. In quantitative terms, the DMI vector can be expressed as

$$\mathbf{D} = D \cdot (\hat{\mathbf{u}}_{ij} \times \hat{\mathbf{z}}),$$

where D is the DMI constant, \mathbf{u}_{ij} represents the unit vector from Fe_i atom to Fe_{ii} atom, and \mathbf{z} represents the unit vector from magnetic Fe_{ii} atom to heavy Te atom. The source of DMI can be considered as follows: (i) the interaction between the Fe_i-Fe_{ii} atom pair and the upper Te atom (corresponding to D_1 vector, represented by the red arrow) and (ii) the interaction between the Fe_i-Fe_{ii} pair and the lower Te atom (corresponding to D_2 vector, represented by the blue arrow). Because the upper and lower triangles share the same Fe_i-Fe_{ii} atom pair, the upper and lower Te atoms adopt the opposite direction for D_1 and D_2 vectors. As for a centrosymmetric structure with Fe_{ii} atom located at the center position of Te-Te atoms, the two vectors thus cancel out, thus vanishing the effective net DMI vector D_{eff} . However, the deviation of Fe_{ii} atoms breaks the inversion symmetry, and thus makes the nonequal D_1 and D_2 yield a nonzero D_{eff} within each monolayer. As for the total DMI (D) of the unit cell, since the D_{eff} vectors in top and bottom layers have the same direction, the magnitude of D is the sum of the two vectors. Moreover, Fig. R11b illustrates the effective net DMI vectors D_{eff} viewed from [0001] zone axis, which are perpendicular to the Fe_i-Fe_{ii}-Te atom cross sections and exhibit threefold rotational symmetry within the ab plane. Furthermore, the DFT calculated value of D for Fe_{2.84±0.05}GaTe₂ sample is 0.91 mJ/m², which is comparable to that extracted from domain width ($D = 0.25$ mJ/m²). This agreement confirms the reliability of our DMI model.

As observed in our previous LTEM and MFM experiments (Fig. 3 and Fig. 4 in the main text), the out-of-plane magnetic field B did not change the shape of skyrmions. The skyrmions remained circular and exhibited a reduction in size as increasing out-of-plane B . In order to further investigate the influence of skyrmion shape with in-plane magnetic field B , we performed micromagnetic simulations in Fig. R12 (Details about the simulation are listed in Fig. R12 Note). It can be seen that as the in-plane magnetic field increases, skyrmions transforms from circular to elliptical at $B = 200$ mT, with the elongated shape along the direction of the in-plane magnetic field.

In the revised manuscript, we have incorporated discussions on formation mechanism of DMI vector D_{eff} from [11 $\bar{2}$ 0] zone axis and [0001] zone axis, respectively (refer to Page 9, Lines 243-246 in the main text and Fig. S11 in Supplementary information). Meanwhile, the simulation of skyrmion-shape variation with in-plane magnetic field B are added in Page 13, Lines 371-374 in the main text and Fig. S21 in Supplementary information.

“Moreover, Fig. S11b illustrates the effective net DMI vectors D_{eff} viewed from

[0001] zone axis, which are perpendicular to the $\text{Fe}_i\text{-Fe}_{ii}\text{-Te}$ atom cross sections and exhibit threefold rotational symmetry within the ab plane.”

“Moreover, our micromagnetic simulations demonstrate the influence of in-plane magnetic field B on skyrmion shape, revealing a progressive transformation from a circular to an elliptical configuration (see **Fig. S21** and **Supplementary Note 7**).”

Fig. R11. **a** Schematic illustration of DMI in asymmetric layers viewed from $[11\bar{2}0]$ zone axis. The red arrow D_1 represents the direction of DMI vector in the upper triangle composed of $\text{Fe}_i\text{-Fe}_{ii}\text{-Te}$, while the blue arrow D_2 represents the lower part in the opposite direction. The black arrow D_{eff} represents the sum of the non-zero DMI vector. **b** Schematic illustration of DMI viewed from $[0001]$ zone axis. The black arrow represents the effective net DMI vectors D_{eff} . And the red rectangular represents the slice of atoms in the left panel **a**.

Fig. R12. Micromagnetic simulations of skyrmion evaluation with in-plane magnetic field B at **a** 0 mT, **b** 100 mT and **c** 200 mT, respectively. The shape of skyrmion transforms gradually from circular to elliptical.

Fig. R12 Note: Micromagnetic simulations were carried out using the GPU-accelerated micromagnetic simulation program Mumax³. Default magnetic parameters used in the simulations include $A = 1.3$ pJ/m, $K_u = 0.8 \times 10^5$ J/m³, $M_s = 2.5 \times 10^5$ A/m, $D = 0.25$ mJ/m², and slab geometries with dimensions of $1024 \times 1024 \times 16$, with a mesh size of $2 \times 2 \times 4$ nm. Periodic boundary conditions were taken into account for large-scale simulations. The initial zero-field skyrmion state was relaxed from a random state with 100 mT magnetic field. Employing the initial skyrmion state as the input, we systematically applied an in-plane magnetic field along y axis, and relaxed the magnetization to a stable state.

Referee B's Comment 4: Why is there such a large variance in the sizes of the skyrmions? this is also different to what is generally reported. In fact, it almost looks bimodal in many images.

Author's reply: We thank the referee's comments on skyrmion size variance. In the LTEM experiments, we initially raised the sample temperature above Curie temperature, then cooled it to the target temperature with a 30 mT external magnetic field, and finally removed the external magnetic field to obtain zero-field skyrmions. As shown in Fig. R13a, the zero-field skyrmion density at low temperature (100 K) was relatively low, exhibiting non-uniformed size distribution. However, at higher temperature 250 K and 320 K, the zero-field skyrmion density gradually increased, and the size distribution became more uniform.

To clarify physical mechanism underlying the variation in skyrmion size at different temperatures, we simulated the zero-field skyrmions after field cooling (see Fig. R13 Note). As is known, the formation of skyrmions is determined by a delicate interplay of the magnetic parameters, including magnetic anisotropy K_u , DMI constant D , saturation magnetization M_s , sample thickness t , and exchange stiffness A . However, our experiments have demonstrated that increasing the sample temperature of Fe_{3-x}GaTe₂ leads to a significant reduction of magnetic anisotropy K_u , while other parameters keep nearly unchanged. Therefore, as shown in Fig. R13b, we decreased magnetic anisotropy constant $K_u = 3.2 \times 10^5$ J/m³, 1.6×10^5 J/m³ and 0.7×10^5 J/m³ in the simulations to represent the increasing of sample temperature. Our simulations demonstrate that for large K_u at low temperature, the density of zero-field skyrmions is low, and the distant between the nearest skyrmions can be considerably large in certain regions, thus facilitating the expansion of skyrmion size upon the removal of the magnetic field. In contrast, with small K_u at high temperature, the skyrmions exhibit a densely hexagonal arrangement to each other, which suppresses the extension of the skyrmions. Consequently, they remain uniformly distributed after removing the

magnetic field.

In summary, we discover that a larger K_u at low temperature results in lower skyrmion density, which exhibit a larger skyrmion-skyrmion distance and allow the expansion of skyrmion size. Consequently, the skyrmion size distribution at lower temperatures is non-uniform. In contrast, at higher temperatures, the lower K_u increases the skyrmion density and reduces the skyrmion-skyrmion distance, leading to a much more uniform size distribution. In the revised manuscript, we have incorporated discussions on skyrmion size variations into main text (refer to Page 13, Lines 358-360) and Supplementary information (refer to Supplementary Note 4 and Fig. S18).

“(see Supplementary **Fig. S17** for detailed size distribution of field-free skyrmions in $\text{Fe}_{2.84\pm 0.05}\text{GaTe}_2$ at varied temperatures, and **Fig. S18** for the corresponding Micromagnetic simulations).”

Fig. R13. **a** Lorentz Phase images of zero-field skyrmion after 30 mT field cooling at 100 K, 250 K and 320 K. **b** Micromagnetic simulations of zero-field skyrmion after 30 mT field cooling with magnetic anisotropy constant $K_u = 3.2 \times 10^5 \text{ J/m}^3$, $1.6 \times 10^5 \text{ J/m}^3$ and $0.7 \times 10^5 \text{ J/m}^3$.

Fig. R13 note: To validate this experimental result, we conducted micromagnetic simulations of the field cooling process and then removing the external magnetic field. Default magnetic parameters used in the simulations include $A = 1.3 \text{ pJ/m}$, $K_u = 0.8 \times 10^5 \text{ J/m}^3$, $M_s = 2.5 \times 10^5 \text{ A/m}$, $D = 0.25 \text{ mJ/m}^2$, and slab geometries with dimensions of $512 \times 512 \times 64$, with a mesh size of $2 \times 2 \times 2 \text{ nm}$. Periodic boundary conditions were taken into account for large-scale simulations. The zero-field skyrmion state was relaxed from a random state with 30 mT magnetic field, and removing the magnetic field to relax until stable.

Referee B’s Comment 5: There needs to be more interpretation of the results and tying back to a structure-property relation for me to be comfortable recommending this paper.

Author's reply: Once again, we express our gratitude to the referees for their insightful suggestions on the manuscript. In an effort to revisit the structure-property relationship, our extended analysis provides a detailed examination of how Fe deficiency disrupts structural symmetry, leading to the displacement of Fe_{ii}, and elucidates its role in the generation of DMI. The integration of theoretical and experimental approaches throughout the entire manuscript establishes a comprehensive discussion. See also author's reply to the Referee B's Comment 2. We hope the referee will be satisfied with the revised manuscript as well as our responses.

Referee B's Comment 6: Smaller notes: You mix cubic and hexagonal coordinates- since the system is hexagonal, you should make this consistent (e.g. [0001] instead of [001], L 138)

Author's reply: Thanks for the referee's suggestion. We have made the necessary modifications to all the hexagonal coordinates.

Referee B's Comment 7: You need to soften some statements in the introduction- e.g. rapid thermal annealing is not going to "revolutionize skyrmion logic" (L103), these processes have been around for a while and, additionally, are not particularly chip compatible.

Author's reply: Thanks for the referee's suggestion. We have revised such expressions as recommended (see Page 4, Lines 104-107).

“More intriguingly, with the use of a homemade in-situ optical LTEM, we realized ultrafast writing of RT skyrmions in the non-stoichiometric Fe_{3-x}GaTe₂ thin flakes by a single femtosecond (fs) laser pulse, which offers a possible avenue for the realization of ultrafast and energy-efficient skyrmion-based logic and memory devices.”

Response to the Report of Referee C

Referee C's General Comment: The authors investigate a non-stoichiometric room temperature magnet $\text{Fe}_{2.86}\text{GaTe}_2$ crystal where the Fe vacancies induce the formation of DMI by spatial inversion symmetry breaking. Such an in-plane isotropic DMI brings about RT Néel-type skyrmions, and the size of the skyrmions can be regulated by the sample thickness and the external magnetic field. The dynamic writing process of RT skyrmions in $\text{Fe}_{2.86}\text{GaTe}_2$ flakes enhances the potential application of spintronic devices.

The paper is timely and of interest.

Author's reply: We sincerely thank the referee for careful reading of our manuscript and pointing out that our paper is “*timely and of interest*”. The further valuable suggestions and comments provided by the referee are greatly helpful to improve our manuscript. Below we answer the comments in a point-by-point basis. We hope the referee will be satisfied with the revised manuscript as well as our responses.

Referee C's Comment 1: The Methods section describes the process for obtaining a $\text{Fe}_{2.86}\text{GaTe}_2$ single crystal, which was achieved directly by precisely controlling the initial molar ratio of the powder mixtures and growing conditions. I am inquiring about the method of determining the optimum molar ratio in this work? by experiment or theoretical calculation. And what is the advantage of using iron deficiency as a means of breaking the centrosymmetric structure compared to other methods, such as elemental doping (Ref. 31)?

Author's reply: We sincerely thank the referee for the valuable comments. In order to control Fe content, we have systematically grown a series of $\text{Fe}_{3-x}\text{GaTe}_2$ single crystals by varying the Fe content in the raw material composition, utilizing a Te-flux method. Subsequently, comprehensive energy-dispersive X-ray spectroscopy (EDS) mapping was conducted on the cleaved surfaces of these crystals to determine their chemical composition. To ensure the reliability of the EDS results, mapping was carried out at four distinct areas for each sample. A comprehensive overview of the raw material composition and final product is outlined in Table R2.

Our experimental results clearly demonstrate that controlling the Fe content in the final crystals is achievable by varying the raw Fe ratio. Specifically, when the raw Fe ratio fall below 0.8, the $\text{Fe}_{3-x}\text{GaTe}_2$ phase cannot be formed. Instead, a mixture of phases, including GaTe and Ga_2Te_3 phases, is produced. In contrast, when the raw Fe ratio is equal to or greater than 0.9, $\text{Fe}_{3-x}\text{GaTe}_2$ single crystals can be crystallized, with the Fe content in these single crystals increasing proportionally with the raw Fe ratio. However,

an increase in the raw Fe ratio to 1.3 results in the formation of FeTe phase.

As surmised in Table R2, the chemical formulas for the crystals with the minimum and maximum Fe content correspond to $\text{Fe}_{2.84\pm 0.05}\text{GaTe}_2$ and $\text{Fe}_{2.96\pm 0.02}\text{GaTe}_2$, respectively. This result indicates that Fe vacancies always exist in the single crystals synthesized using a Te-flux method. In the previous version of our manuscript, to highlight the existence of Fe vacancies, we reported the observation of Néel-type skyrmions in the $\text{Fe}_{3-x}\text{GaTe}_2$ single crystals synthesized with a raw Fe ratio of 0.9. This ratio represents the minimum Fe content for crystalline iron-deficient samples.

In the revised manuscript, to enhance the accuracy of the chemical formula, an error bar has been added by summarizing the EDS results obtained at different areas, and the chemical formula is denoted as $\text{Fe}_{2.84\pm 0.05}\text{GaTe}_2$. We have incorporated discussions on controlling the Fe content of the crystals into both the main text (refer to Page 5, Lines 110-121), Method section (refer to Page 17, Lines 498-500) and Supplementary information (refer to Supplementary Note 1 and Table S1), providing a more comprehensive understanding of the growth conditions and their impact on the final composition of the single crystals.

We would like to emphasize that both elemental doping and iron deficiency have been experimentally demonstrated as efficient means of breaking the centrosymmetric structure. In contrast to the elemental doping [**Science Advances** **8.12 (2022): eabm7103**], the advantage of using iron deficiency lies in the ease of sample preparation. As surmised in Table R2, Fe vacancies are consistently present in the single crystals synthesized with a raw Fe ratio from 0.9 to 1.2. As for $(\text{Fe}_y\text{Co}_{1-y})_5\text{GeTe}_2$ compounds grown by chemical vapor transport, the structure and magnetism are highly sensitive to composition. Only samples with a doping Co content precisely at 50% exhibit a polar structure and room-temperature ferromagnetism. When the doping Co content is 47% or lower, the samples adopt a centrosymmetric structure and antiferromagnetic properties [**Physical Review Materials** **4.7 (2020): 074008**]. In the revised manuscript, the associated discussions on the advantage of using iron deficiency were added (see Page 5, lines 117-121 in the main text), as also shown below.

“In order to control the Fe content, we systematically grew a series of $\text{Fe}_{3-x}\text{GaTe}_2$ single crystals by varying the Fe content in the raw material composition, utilizing a Te-flux method (see **Methods** section and **Supplementary Note 1**). To determine the chemical composition of the as-grown crystals, energy dispersive X-ray spectroscopy (EDX) analyses were conducted on the surfaces of $\text{Fe}_{3-x}\text{GaTe}_2$ nanoflakes (**Fig. 1a**) that were exfoliated and placed onto the Si_3N_4 membrane (see **Methods**). The ratio of raw

materials and the corresponding final crystal composition are listed in **Table S1**, Supplementary **Fig. S1** and **Fig. 1b**. We found that the Fe deficiencies always exist in these crystals, while the minimum and maximum Fe contents correspond to $\text{Fe}_{2.84\pm 0.05}\text{GaTe}_2$ and $\text{Fe}_{2.96\pm 0.02}\text{GaTe}_2$, respectively. This result implies the feasibility of inducing Fe deficiency in the samples. To highlight the existence of Fe vacancies, the subsequent studies were focused on the minimum Fe content sample $\text{Fe}_{2.84\pm 0.05}\text{GaTe}_2$.”

Table R3. Summary of the raw material composition and the final product for the growth of $\text{Fe}_{3-x}\text{GaTe}_2$ samples using the self-flux method.

Molar ratio of Fe: Ga: Te	Mass of Fe (g)	Mass of Ga (g)	Mass of Te (g)	Product
0.6:1:2	0.9349	1.9452	7.1200	GaTe, Ga ₂ Te ₃
0.7:1:2	1.0739	1.9153	7.0107	
0.8:1:2	1.2088	1.8864	6.9048	
0.9:1:2	1.3397	1.8583	6.8020	$\text{Fe}_{2.84\pm 0.05}\text{GaTe}_2$
1.0:1:2	1.4667	1.8310	6.7023	$\text{Fe}_{2.91\pm 0.04}\text{GaTe}_2$
1.1:1:2	1.5900	1.8046	6.6054	$\text{Fe}_{2.95\pm 0.03}\text{GaTe}_2$
1.2:1:2	1.7099	1.7789	6.5113	$\text{Fe}_{2.96\pm 0.02}\text{GaTe}_2$
1.3:1:2	1.8263	1.7539	6.4198	FeTe

Referee C’s Comment 2: Please check Supplementary Note 1 and Table S1 to verify the value of non-stoichiometric $\text{Fe}_{3-x}\text{GaTe}_2$.

Author’s reply: We sincerely thank the referee for the valuable comments. In the previous version of our manuscript, we reported the observation of Néel-type skyrmions in the $\text{Fe}_{3-x}\text{GaTe}_2$ single crystals synthesized with a raw Fe ratio of 0.9. Their average chemical formula was denoted as $\text{Fe}_{2.86}\text{GaTe}_2$ for EDS mapping while $\text{Fe}_{2.79}\text{GaTe}_2$ for XRD refinement. It is crucial to highlight that all the crystals were grown using the same temperature conditions and raw material composition. The observed variation in the average chemical formula can be attributed to measurement errors associated with different characterization techniques.

In the revised manuscript, to enhance the accuracy of the chemical formula, an error bar has been added by summarizing the EDS results obtained at four different areas, and the chemical formula is denoted as $\text{Fe}_{2.84\pm 0.05}\text{GaTe}_2$, which remains within the error range of the XRD refined chemical formula $\text{Fe}_{2.79}\text{GaTe}_2$. We have integrated error analysis of EDS determined chemical formula into both the main text (refer to Page 5, Lines 118-119) and Supplementary information (refer to Supplementary Note 1 and Table S1).

Referee C’s Comment 3: “Moreover, the size of the skyrmions decreases as the sample

thickness becomes thinner, and field-free sub-100 nm skyrmions can be obtained at RT when the thickness falls below a threshold ranging from 40 to 60 nm.” This value is significantly lower compared to other vdW magnets based on iron, as summarised in Fig. 4c. This could be attributed to the competition among DMI, magnetic dipolar interaction, and magnetic anisotropy. Could you please provide further explanation for the presence of sub-100 nm RT skyrmions in $\text{Fe}_{2.86}\text{GaTe}_2$ nanoflakes?

Author’s reply: We appreciate the referee's comments concerning the physics underlying the skyrmion size. It is widely recognized that the skyrmion size is greatly influenced by the interplay among various factors, including DMI, magnetic dipolar interaction, magnetic exchange interaction, saturation magnetization, and magnetic anisotropy. Utilizing the experimentally established magnetic parameters associated with these magnetic interactions in different vdW magnets (Table R3), we conducted micromagnetic simulations to clarify the contributions of these factors to the skyrmion size (Fig. R14). Further details regarding the simulations are comprehensively described in the Fig. R14 note.

In our initial simulation, we modeled the zero-field skyrmion state following a 60 mT field cooling, adopting the magnetic parameters of $(\text{Fe}_{0.5}\text{Co}_{0.5})_5\text{GeTe}_2$ as a reference [Science Advances 8.12 (2022): eabm7103]. The simulation results, as depicted in Fig. R14, revealed a high-density of skyrmions with an average size of approximately 116 nm. Subsequently, by initiating the simulation with this skyrmion state and progressively reducing the DMI constant D to match that of $\text{Fe}_{3-x}\text{GaTe}_2$ while keeping other magnetic parameters constant (Fig. R14a and R14b), we observed a significant decrease in the skyrmion size to 81 nm at $D = 0.25 \text{ mJ/m}^2$. Employing a similar approach, we further simulated the magnetic domain states by varying the saturation magnetization M_s , sample thickness t , magnetic anisotropy constant K_u and exchange stiffness A towards those of $\text{Fe}_{3-x}\text{GaTe}_2$. It is clearly demonstrated that each parameter reduction leads to a decrement in the skyrmion size. Thus, our simulations suggest that smaller magnetic parameters such as D , M_s , t , K_u , and A in $\text{Fe}_{3-x}\text{GaTe}_2$ (in comparison to $(\text{Fe}_{0.5}\text{Co}_{0.5})_5\text{GeTe}_2$), contribute to the reduction in skyrmion size.

In the revised manuscript, we have incorporated discussions on micromagnetic simulations to provide further explanation for the presence of sub-100 nm RT skyrmions in $\text{Fe}_{2.84\pm 0.05}\text{GaTe}_2$ nanoflakes. Please refer to the main text (refer to Page 13, Lines 366-372, as also shown below) and Supplementary information (refer to Supplementary Note 6, Table S3 and Fig. S20).

“Compared with the previously reported skyrmion-hosting 2D material

(Fe_{0.5}Co_{0.5})₅GeTe₂³¹, Fe_{2.84±0.05}GaTe₂ exhibits smaller magnetic parameters such as DMI constant D , saturation magnetization M_s , threshold of sample thickness t and etc. (see **Fig. S19** and **Supplementary Note 5** for the determination of magnetic parameters), which contribute to the reduction in skyrmion size (see **Table S3**, **Fig. S20** and **Supplementary Note 6** for the corresponding micromagnetic simulations).”

Table R4. Magnetic parameters for skyrmion-host 2D materials.

	(Fe _{0.5} Co _{0.5}) ₅ GeTe ₂	Fe _{3-x} GaTe ₂
D (mJ/m ²)	0.90	0.25
M_s ($\times 10^5$ A/m)	3.0	2.5
t (nm)	≥ 110	≥ 46
K_u ($\times 10^5$ J/m ³)	2.4	0.8
A (pJ/m)	4.0	1.3
T (K)	300	300

Fig. R14. Variation of simulated magnetic structure and corresponding skyrmion sizes with varied **a, b** DMI constant D , **c, d** saturation magnetization M_s , **e, f** sample thickness t , **g, h** magnetic anisotropy constant K_u , and **i, j** exchange stiffness A .

Fig. R14 note: Micromagnetic simulations were carried out using the GPU-accelerated micromagnetic simulation program Mumax³. Unless specified otherwise, default magnetic parameters used in the simulations include $A = 4.0$ pJ/m, $K_u = 2.4 \times 10^5$ J/m³, $M_s = 3.0 \times 10^5$ A/m, $D = 0.90$ mJ/m², and slab geometries with dimensions of $512 \times 512 \times 64$, with a mesh size of $2 \times 2 \times 2$ nm. Periodic boundary conditions were taken into account for large-scale simulations. The initial skyrmion state was relaxed from a random state with 60 mT magnetic field. Employing the initial skyrmion state as the input, we systematically varied the magnetic parameters— D , M_s , t , K_u , and A individually—subsequently allowing the magnetization to evolve into a stabilized state through relaxation processes.

Referee C's Comment 4: The authors should stress what is the difference between the Fe_i and Fe_{ii}, as this will contribute to understanding why the deviation happened on the Fe_{ii} atoms.

Author's reply: We highly appreciate the referee's comments, which are greatly helpful for us to improve the quality of our manuscript. After a reanalysis of STEM images and a careful examination of the XRD data, we have made new discoveries regarding the Fe_i and Fe_{ii} vacancies. Integrated with first-principles calculations, we have presented a more comprehensive discussion on Fe_{ii} deviations.

In order to provide a clearer view of the Fe_i and Fe_{ii} columns, we acquired improved ABF-STEM images of the Fe_{2.84±0.05}GaTe₂ sample along the $[11\bar{2}0]$ zone axis. As depicted in Fig. R15a, the magnified ABF-STEM image of the single layer distinctly shows the Fe_i and Fe_{ii} atoms adjacent to the Ga atoms. For a quantitative determination of the deviation of the Fe_{ii} atoms, we focused on the region marked by the left blue rectangle comprising the Te-Fe_{ii}-Te atoms in Fig. R15a. We then vertically integrated the corresponding imaging intensity line profile (Fig. R15b). By referencing the center of the two Te atoms, the deviation of the Fe_{ii} atom towards the $-c$ direction was determined to be -0.20 Å. Utilizing the same procedure, we surveyed an area of 2×17 unit cells, yielding an average Fe_{ii} deviation of -0.16 ± 0.06 Å.

Additionally, we observed that the image intensity of Fe_{i-a} above Fe_{ii} is weaker than that of Fe_{i-b} below Fe_{ii}, as evident in the imaging intensity line profile of Fe_{i-a}-Fe_{ii}-Fe_{i-b} atoms indicated by the right blue rectangle in Fig. R15c. Since imaging intensity is generally proportional to the number of projected atoms [PNAS **107.26 (2010): 11682-11685**], the contrast difference between Fe_{i-a} and Fe_{i-b} indicates asymmetric site occupations, suggesting a small quantity of Fe vacancies in the Fe_{i-a} site. In the previous version of our manuscript, we emphasized the Fe deficiency at Fe_{ii} sites with an occupancy ratio of 0.8467. Upon re-evaluating the results of the refined single-crystal

XRD, we discovered that the lower Fe_{i-b} sites are nearly fully occupied, while the higher Fe_{i-a} sites have an occupancy ratio of 0.9688. Thus, we claim that the Fe deficiency occurs not only at those Fe_{ii} positions, but also happens in Fe_{i-a} positions.

Having established the crystal structure, we conducted first-principles calculations to analyze the impact of Fe_{i-a} and Fe_{ii} vacancies on Fe_{ii} deviation. Starting with a perfect centrosymmetric Fe_3GaTe_2 lattice, we constructed a 2×2 supercell encompassing a total of four molecular layers. Within the bottommost layer, we systematically introduced three scenarios: no vacancy, Fe_{i-a} vacancy, and Fe_{ii} vacancy. Lattice relaxations were performed independently for the three distinct supercells. In order to illustrate the Fe vacancies and further compare the alterations in Fe_{ii} chemical bonding, we presented the electron density of Fe_{3-x}Ga atoms within the bottommost layer (Fig. R16). Specific computational details can be found in Fig. R16 note.

Fig. R16a and R16b illustrate the scenario with no vacancy. The electron density (colored in yellow) strongly overlaps between Fe_{ii} -Ga atoms, forming the Fe_{ii} -Ga honeycomb lattice plane with robust chemical bonding (highlighted by black dashed lines). Simultaneously, the Fe_{i-a} and Fe_{i-b} dimers are located at the center of the Fe_{ii} -Ga honeycomb lattice but do not bond with Fe_{ii} -Ga atoms. Consequently, the chemical bonding is mirror-symmetric along the Fe_{ii} -Ga plane, leading to the absence of Fe_{ii} displacements. Thus, perfect Fe_3GaTe_2 exhibits a centrosymmetric crystal structure with $c \rightarrow -c$ mirror symmetry.

Fig. R16c and R16d illustrate the scenario with Fe_{ii} vacancy. Although Fe_{ii} vacancies cause deformation of the Ga electron density within the ab plane, there is still no overlapping with of Fe_{i-a} and Fe_{i-b} in the c direction. This observation indicates that the remaining Fe_{ii} primarily forms bonds with Ga in the ab plane, with no displacement observed in the c direction.

Fig. R16e and R16f illustrate the scenario with Fe_{i-a} vacancy. Apart from the strongly bonded Fe_{ii} -Ga honeycomb lattice, there is additional electron-density overlapping among the lower Fe_{i-b} atom and its three nearest Fe_{ii} atoms, while no overlapping between Fe_{i-b} and its three nearest Ga atoms. Consequently, the newly formed chemical bonding (highlighted by black dashed line) will drag the Fe_{ii} atoms displacing downwards to the lower Fe_{i-b} atom. Notably, the determined deviation between nearest Fe_{ii} and Ga atoms in the relaxed structure is -0.0554 \AA . This value aligns with the analysis from ABF-STEM image, confirming the reliability of our DFT model. Based on the DFT results above (Fig. R16), we can conclude that the asymmetric vacancy of Fe_{i-a} induces a displacement of Fe_{ii} atoms towards the $-c$

direction, resulting in the spatial inversion symmetry breaking.

In addition, we calculated the vacancy formation energy for Fe_i and Fe_{ii} . As for the calculation, we used a structure that repeats 3 times in the c direction and removed the bottommost Fe_{i-a} or Fe_{ii} atoms, and compared its total free energy with the parent structure. The calculated values of formation energy for Fe_i and Fe_{ii} was 2.96 eV/Fe and 2.86 eV/Fe, respectively. This result suggests that Fe_{ii} vacancies are more likely to occur, in line with the experimental result.

In the revised version of the manuscript, we have integrated experimental findings and DFT calculations on Fe_{ii} deviations, Fe_{ii} vacancies and the asymmetric Fe_i vacancies (refer to Page 6, Lines 154-159, Page 7 and 8, Lines 187-219 in the main text), as also shown below. The corresponding analyses and experimental data are comprehensively presented in Supplementary Note 2, Note 3, Fig. S5 and Fig. S10.

“However, the HAADF image along the $[11\bar{2}0]$ zone axis (**Fig. 1f**) reveals that Fe_{ii} atoms deviate clearly from the center position of the Te slices along the c -axis, which is also supported by the annular bright-field (ABF-STEM) image in **Fig. S4**. By referencing the center of the two Te atoms in a magnified ABF-STEM image (**Fig. S5**), an averaged Fe_{ii} deviation is calculated as $-0.16 \pm 0.06 \text{ \AA}$ over an area of 2×17 unit cells (see Supplementary **Note 2**).”

“In comparison to the stoichiometric Fe_3GaTe_2 with a centrosymmetric structure, the presence of Fe deficiency in $\text{Fe}_{2.84 \pm 0.05}\text{GaTe}_2$ should exert a pivotal influence on the Fe_{ii} deviation for the asymmetric structure. Our refined single-crystal XRD indicates that Fe deficiency is predominantly concentrated at the Fe_{ii} positions with an occupancy ratio of 0.8467. Additionally, the upper-layer Fe_{i-a} sites have an occupancy ratio of 0.9688, while the under-layer Fe_{i-b} sites are nearly fully occupied (**Fig. 1h**). As observed in the line profile of Fe_{i-a} and Fe_{i-b} atoms in the ABF-STEM image (**Fig. S5c**), it is apparent that the image intensity of Fe_{i-a} above Fe_{ii} is weaker than that of Fe_{i-b} below Fe_{ii} . Since the ABF imaging intensity is generally proportional to the number of projected atoms⁴¹, the contrast difference between Fe_{i-a} and Fe_{i-b} indicates asymmetric site occupations, suggesting a small quantity of Fe vacancies in the Fe_{i-a} site, which is consistent with the results of single-crystal XRD. To assess the influence of Fe_{i-a} and Fe_{ii} vacancies on Fe_{ii} deviation, we further conducted first-principles calculations involving structure relaxation under three scenarios: no vacancy, Fe_{i-a} vacancy, and Fe_{ii} vacancy (Supplementary **Note 3**). The electron density of Fe_{3-x}Ga atoms is depicted in **Fig. S10** to facilitate a comparison of the alterations in Fe_{ii} chemical bonding: (a) The

perfect Fe_3GaTe_2 , with no Fe vacancy, showcases a hexagonally bonded $\text{Fe}_{\text{ii}}\text{-Ga}$ plane. In this arrangement, the centrally positioned $\text{Fe}_{\text{i-a}}$ and $\text{Fe}_{\text{i-b}}$ dimers do not form direct bonds with Fe_{ii} and Ga atoms. Thus, the overall chemical bonding is mirror-symmetric along the $\text{Fe}_{\text{ii}}\text{-Ga}$ plane with no Fe_{ii} deviation. (b) The presence of Fe_{ii} vacancy induces deformation of the Ga electron density within the ab plane. Nevertheless, no bonding is established between the $\text{Fe}_{\text{ii}}\text{-Ga}$ plane and the Fe_{i} atoms, which remain a mirror-symmetric electron density with no Fe_{ii} deviation. (c) In case of $\text{Fe}_{\text{i-a}}$ vacancy, there is additional electron-density overlapping between the lower $\text{Fe}_{\text{i-b}}$ atom and its three nearest Fe_{ii} atoms, while no overlapping between $\text{Fe}_{\text{i-b}}$ and its three nearest Ga atoms. As a result, chemical bonding between the Fe_{ii} and $\text{Fe}_{\text{i-b}}$ atoms induces a substantial Fe_{ii} deviation, with a calculated $\delta c(\text{Fe}_{\text{ii}} - \text{Ga})$ of about -0.0554 \AA , which compares favorably to the XRD result of -0.0871 \AA . Furthermore, the calculated formation energy value for Fe_{i} vacancy (2.96 eV/Fe) is higher than Fe_{ii} vacancy (2.86 eV/Fe), indicating that the formation of Fe_{ii} vacancies is more favorable. The above Fe vacancy model and calculated Fe_{ii} deviation align well with the analysis from single-crystal XRD and ABF-STEM image. Therefore, we conclude that the asymmetric vacancy of $\text{Fe}_{\text{i-a}}$ induces a displacement of Fe_{ii} atoms towards the $-c$ direction, which results in the symmetry breaking of the $\text{Fe}_{2.84\pm 0.05}\text{GaTe}_2$ crystal structure.”

Fig. R15. a Magnified ABF-STEM image of the single $\text{Fe}_{2.84(5)}\text{GaTe}_2$ layer. **b** Integrated imaging intensity line profile along the c -axis within the area marked by the $\text{Te-Fe}_{\text{ii}}\text{-Te}$ atoms in the blue rectangles. The red region indicates the Fe_{ii} deviation from the centers

of Te-Te atoms. **c** Integrated imaging intensity line profile of Fe_{i-a}-Fe_{i-b} atoms.

Fig. R16. The relaxed crystal structure and corresponding electron density of Fe_{3-x}Ga atoms sliced from Fe_{3-x}GaTe₂ with **a, b** no vacancy, **c, d** Fe_{ii} vacancy and **e, f** Fe_{i-a} vacancy. The black dashed line indicates the chemical bonding with electron-density overlapping. The yellow-colored electron densities are shown at the same isosurface value.

Fig. R16 note: In all cases presented, the Vienna ab initio simulation package (VASP) was used with electron-core interactions described by the projector augmented wave method for the pseudopotentials, and the exchange correlation energy calculated with the generalized gradient approximation of the Perdew-Burke-Ernzerhof (PBE) form. The plane wave cutoff energy was 400 eV for all the calculations. In calculating the atomic shifts due to Fe_{i-a} and Fe_{ii} vacancies, we used a 2×2 supercell and removed the bottommost Fe_{i-a} and Fe_{ii} atoms. The Monkhorst-Pack scheme was used for the Γ -centred $12 \times 12 \times 1$ k-point sampling. All atoms' relaxations were performed until the force become smaller than 0.001 eV/Å for determining the most stable geometries.

Referee C's Comment 5: Could the authors present the X-ray diffraction pattern of the Fe_{2.86}GaTe₂ single crystal?

Author's reply: We appreciate the suggestions from the referee. In response, we have reanalyzed the single-crystal XRD data of Fe_{2.84±0.05}GaTe₂, and provided the synthetic

XRD patterns for $(hki0)$, $(h0\bar{h}l)$, and $(0k\bar{k}l)$ reflections in Fig. R17.

As shown in Fig. R17a and R17b, both XRD patterns exhibit a series of $(000l)$ reflections, such as $(000\bar{5})$ and (0003) . Generally, the odd l ($l = 2n + 1$) values of $(000l)$ and $(h\bar{h}2\bar{h}l)$ reflections are allowed for the non-centrosymmetric space group $P3m1$, but forbidden in a centrosymmetric space group $P6_3/mmc$. The XRD refined non-centrosymmetric structure of $\text{Fe}_{2.84\pm 0.05}\text{GaTe}_2$ (space group $P3m1$, with Fe_{ii} deviation) align well with the symmetry-allowed reflections. Thus, the presence of odd l values in XRD reflections, such as (0003) and $(000\bar{5})$, serves as additional confirmation that the space group of $\text{Fe}_{2.84\pm 0.05}\text{GaTe}_2$ is non-centrosymmetric $P3m1$, rather than the centrosymmetric $P6_3/mmc$. In the revised manuscript, we incorporated discussions on XRD analysis in both the main text (refer to Page 6 and 7, Lines 169-172) and Supplementary information (refer to Fig. S6).

“In the case of $\text{Fe}_{2.84\pm 0.05}\text{GaTe}_2$, however, a series of weak $(11\bar{2}l)$ and $(000l)$ reflection patterns, such as $(11\bar{2}3)$ and $(11\bar{2}5)$ (see Fig. 1g and Fig. S6), were detected for $l = 2n + 1$, suggesting a substantial deviation from the original crystal structure of Fe_3GaTe_2 .”

Fig. R17 The synthetic X-ray diffraction patterns of $\text{Fe}_{2.84\pm 0.05}\text{GaTe}_2$ single crystal for **a** $(hki0)$, **b** $(h0\bar{h}l)$, and **c** $(0k\bar{k}l)$ reflections.

Referee C’s Comment 6: For most of iron-based ternary tellurides, nanoflakes tend to degrade easily in air. How did the authors avoid degradation during the whole measurements?

Author’s reply: We sincerely thank the referee for the valuable comments. Drawing upon our experience, thin-layered samples (~ 10 nm) of iron-based ternary tellurides tend to degrade within about 30 minutes when exposed to air, whereas thicker samples (~ 100 nm in thickness) demonstrate enhanced stability and can endure for an extended duration. To prevent degradation during the measurement process, we implemented the following methods:

The fresh $\text{Fe}_{2.84\pm 0.05}\text{GaTe}_2$ nanoflakes utilized for Hall devices, LTEM and MFM experiments were prepared through an all-dry mechanical-transfer method within an argon-filled glovebox. In an argon-protected environment, the nanoflakes were first produced on PDMS stamp by micromechanical cleavage, and then transferred onto Si_3N_4 membrane or SiO_2/Si substrate, with or without pre-patterned Au electrodes. To maintain the integrity of the samples, these freshly prepared Hall devices, LTEM, and MFM specimens were promptly placed into a plastic box within the argon-filled glovebox and securely sealed with parafilm before removal. During subsequent measurements outside the glovebox, the exposure of the samples to air was minimized, and all the sample transfers were conducted within a confined timeframe of no more than 10 minutes. Hall and LTEM measurements were conducted under vacuum conditions, while MFM measurements were carried out in an environment continuously flushed with argon gas. These methods ensure the effective protection of the samples throughout the testing procedures.

In the revised manuscript, we have added more descriptions on sample preparation. (see Method section in Page 18, Lines 508-518 and 529-532).

Referee C's Comment 7: The authors should strive for consistency across various measurements. A sample with a thickness of 250 nm was selected for studying magneto-transport signatures, while sub-100 nm RT skyrmions were observed in a thickness range of 40 to 60 nm.

Author's reply: We appreciate the referee's suggestions regarding the sample thickness. In addition to the magneto-transport measurements for the 250 nm thickness sample, we have further supplemented magnetic field-dependent Hall resistivity ρ_{xy} and topological Hall resistivity ρ_{xy}^T for samples with thicknesses of 110 nm, 85 nm, 70 nm, and 40 nm under room temperature (RT), as illustrated in Fig. R18.

As magnetic field sweeps, Hall resistivity ρ_{xy} for all the samples exhibits nonlinear variation, from which topological Hall signals can be extracted. Moreover, as the sample thickness increases, topological Hall signals gradually strengthen and shift towards higher magnetic field. These observations align with the thickness dependent skyrmion phase diagram (Fig. 4 in the main text). Thus, our findings demonstrate the consistency of the RT topological Hall effect and the presence of RT skyrmions across the entire thickness range, ranging from 40 nm to 250 nm (refer to Fig. 4 in the main text).

In the earlier version of our manuscript, to highlight the presence of topological Hall signals for better clarity, we chose to present magneto-transport measurements

specifically for the 250 nm sample thickness. In the revised manuscript, to strive for consistency of sample thickness, we have supplemented discussion on thickness dependent RT Hall resistivity ρ_{xy} and topological Hall resistivity ρ_{xy}^T in both the main text (refer to Page 10, Lines 281-286, as also shown below) and Supplementary information (refer to Fig. S13).

“Furthermore, we investigated thickness- and magnetic field-dependent Hall resistivity ρ_{xy} and topological Hall resistivity ρ_{xy}^T under room temperature (Fig. S13). As the sample thickness increases, the topological Hall signals gradually strengthen and shift towards higher magnetic field. More importantly, the THE signals persist over a broad temperature range and various thickness, suggesting the existence of topological spin configurations in $\text{Fe}_{2.84\pm 0.05}\text{GaTe}_2$.”

Fig. R18. a-d Room-temperature magnetic hysteresis of Hall resistivity ρ_{xy} and topological Hall resistivity ρ_{xy}^T at various sample thickness from 40 nm to 110 nm. Red (blue) curves were measured with increasing (decreasing) magnetic field. The insets show the optical image of $\text{Fe}_{2.84\pm 0.05}\text{GaTe}_2$ Hall devices.

Referee C’s Comment 8: The manuscript should be improved by a careful reading. There are several corrections to be made. For example:

- Just below Fig. 2: ‘from +6T to 6T’
- Below Fig. S8: ‘ Sample Morphology’

Author’s reply: We sincerely thank the referee for the valuable comments. We have revised such expressions as recommended.

REVIEWERS' COMMENTS

Reviewer #1 (Remarks to the Author):

The authors have provided additional data to support the determination of the new phase. I am satisfied with the response, and I would recommend to proceed to publication. Just a note to the authors, it's easier to use CBED whole pattern and look into the HOLZ lines to check the symmetry.

Reviewer #2 (Remarks to the Author):

Reviewer comments on "Room-temperature sub-100 nm Néel-type skyrmions in non-stoichiometric van der Waals ferromagnet $\text{Fe}_{3-x}\text{GaTe}_2$ with ultrafast laser writability"

The authors have clearly spent time answering my and the other reviewer's concerns and with the clarity that has been added to the introduction, namely the larger emphasis on the role of vacancy ordering, I am much more comfortable recommending this work for publication.

I think there are a few textual things that could be changed to increase the impact of the work, but the story and the physics are much clearer in the newer version.

1. I think a subset of the SEAD patterns from Fig R2 should be added to Figure 1- the superlattice peaks are the strongest evidence for ordering and help explain the DMI. This could replace Fig 1g, since it is the same information but clearer.

2. I think there can be more explicit attention in the text to how pulsed writing is different from the quasi-thermodynamic state. E.g., as in the response:

"Typically, without a magnetic field, zero-field cooling can only result in the formation of interconnected, relatively long stripe domains, but does not spontaneously lead to the creation of skyrmions (as show in Fig. R5a-c) [Nature Communications 13.1 (2022): 3035]. "

Could be added directly to the text.

3. The authors could also make the statement "spin clusters that contain topological defects such as skyrmionic and anti-skyrmionic nucleation centers (snapshot at t_1) due to the ultrafast cooling at a quenching rate of up to 10^{12} K/s" (L457) more obvious to make the work read better.

Overall, the authors have rigorously answered the concerns in my initial review and I recommend the manuscript for publication.

Reviewer #3 (Remarks to the Author):

In the revised manuscript and supplementary information, the authors have thoroughly

characterised their sample, including EDS mapping and XRD data. In addition, they have made new discoveries regarding the Fe_i and Fe_{ii} vacancies and provide a reasonable explanation for the symmetry breaking of the Fe_{2.84±0.05}GaTe₂ crystal structure, which is also consistent with first-principles calculations. Overall, the authors have adequately addressed the reviewers' comments. I recommend the manuscript for publication in *Nature Communication*.

Response to the Report of Referee A

Referee A's General Comment: The authors have provided additional data to support the determination of the new phase. I am satisfied with the response, and I would recommend to proceed to publication. Just a note to the authors, it's easier to use CBED whole pattern and look into the HOLZ lines to check the symmetry.

Author's reply: We sincerely thank the reviewer for recommending our manuscript to be published in *Nature Communications*. The valuable suggestions and comments furnished by the referee have significantly contributed to enhancing the quality of our manuscript.

Response to the Report of Referee B

Referee B's General Comment: The authors have clearly spent time answering my and the other reviewer's concerns and with the clarity that has been added to the introduction, namely the larger emphasis on the role of vacancy ordering, I am much more comfortable recommending this work for publication.

Author's reply: We sincerely thank the reviewer for recommending our manuscript to be published in *Nature Communications*.

Referee B's Comment 1: I think there are a few textual things that could be changed to increase the impact of the work, but the story and the physics are much clearer in the newer version.

I think a subset of the SEAD patterns from Fig R2 should be added to Figure 1- the superlattice peaks are the strongest evidence for ordering and help explain the DMI. This could replace Fig 1g, since it is the same information but clearer.

Author's reply: We sincerely thank the reviewer for the valuable suggestions. We have replaced the figures as recommended.

Referee B's Comment 2: I think there can be more explicit attention in the text to how pulsed writing is different from the quasi-thermodynamic state. E.g., as in the response: "Typically, without a magnetic field, zero-field cooling can only result in the formation

of interconnected, relatively long stripe domains, but does not spontaneously lead to the creation of skyrmions (as show in Fig. R5a-c) [Nature Communications 13.1 (2022): 3035]. ” Could be added directly to the text.

Author’s reply: We sincerely thank the reviewer for the valuable suggestions. We have added the discussions as recommended.

“Furthermore, conventional zero-field cooling can only result in the formation of interconnected, relatively long stripe domains, but does not spontaneously lead to the creation of skyrmions. To demonstrate the differences with in-site fs laser quenching approach, we conducted fluence-dependent laser pulse excitation without magnetic field.”

Referee B’s Comment 3: The authors could also make the statement “spin clusters that contain topological defects such as skyrmionic and anti-skyrmionic nucleation centers (snapshot at t_1) due to the ultrafast cooling at a quenching rate of up to 10^{12} K/s” (L457) more obvious to make the work read better.

Author’s reply: We sincerely thank the reviewer for the valuable suggestions. We have revised the statements as recommended.

“As shown in Fig. 5d (see details in Movie S1), following the excitation by the femtosecond (fs) laser pulse, the initial melted spin state (snapshot at t_0) rapidly evolves into numerous nanoscale spin clusters. These clusters contain topological defects, including skyrmionic and anti-skyrmionic nucleation centers (snapshot at t_1). This transformation occurs due to the ultrafast cooling, achieved at a quenching rate of up to 10^{12} K/s.”

Overall, the authors have rigorously answered the concerns in my initial review and I recommend the manuscript for publication.

Author’s reply: We sincerely thank the reviewer for recommending our manuscript to be published in *Nature Communications*.

Response to the Report of Referee C

Referee C's General Comment: In the revised manuscript and supplementary information, the authors have thoroughly characterised their sample, including EDS mapping and XRD data. In addition, they have made new discoveries regarding the Fe_i and Fe_{ii} vacancies and provide a reasonable explanation for the symmetry breaking of the Fe_{2.84±0.05}GaTe₂ crystal structure, which is also consistent with first-principles calculations. Overall, the authors have adequately addressed the reviewers' comments. I recommend the manuscript for publication in Nature Communication.

Author's reply: We sincerely thank the reviewer for recommending our manuscript to be published in *Nature Communications*. The valuable suggestions and comments furnished by the referee are greatly helpful to improve our manuscript.